# *ExpProof* : Operationalizing Explanations for Confidential Models with ZKPs

**Chhavi Yadav** [* 1]  **Evan Monroe Laufer** [* 2]  **Dan Boneh** [2]  **Kamalika Chaudhuri** [1]

## Abstract

In principle, explanations are intended as a way to increase trust in machine learning models and are often obligated by regulations. However, many circumstances where these are demanded are adversarial in nature, meaning the involved parties have misaligned interests and are incentivized to manipulate explanations for their purpose. As a result, explainability methods fail to be operational in such settings despite the demand (Bordt et al., 2022). In this paper, we take a step towards operationalizing explanations in adversarial scenarios with Zero-Knowledge Proofs (ZKPs), a cryptographic primitive. Specifically we explore ZKP-amenable versions of the popular explainability algorithm LIME and evaluate their performance on Neural Networks and Random Forests. Our code is publicly available at : https://github.com/emlaufer/ExpProof.

## 1. Introduction

> "Bottom line: Post-hoc explanations are highly problematic in an adversarial context" (Bordt et al., 2022)

Explanations have been seen as a way to enhance trust in machine learning (ML) models by virtue of making them transparent. Although starting out as a debugging tool, they are now also widely proposed to prove fairness and sensibility of ML-based predictions for societal applications, in research studies (Langer et al., 2021; Smuha, 2019; Kästner et al., 2021; Von Eschenbach, 2021; Leben, 2023; Karimi et al., 2022; Wachter et al., 2017; Liao & Varshney, 2021) and regulations alike (Right to Explanation (Wikipedia contributors, 2025)). However, as discussed in detail by (Bordt et al., 2022), many of these use-cases are adversarial in nature where the involved parties have misaligned interests

---

[*]Equal contribution  [1]UC San Diego [2]Stanford University. Correspondence to: Chhavi Yadav <chhaviyadav123@gmail.com>, Evan Monroe Laufer <emlaufer@stanford.edu>.

*Proceedings of the $42^{nd}$ International Conference on Machine Learning*, Vancouver, Canada. PMLR 267, 2025. Copyright 2025 by the author(s).

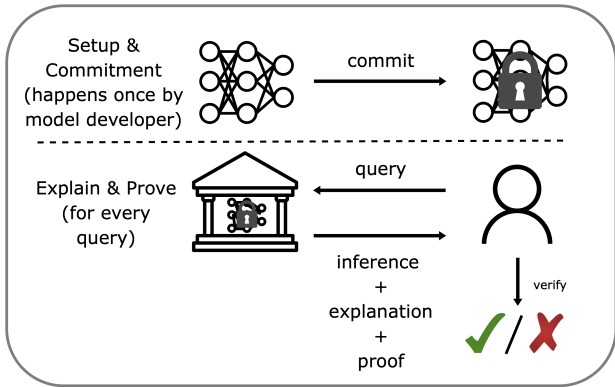

*Figure 1.* Pictorial Representation of *ExpProof*.

and are incentivized to manipulate explanations to meet their ends. For instance, a bank which denies loan to an applicant based on an ML model's prediction has an incentive to return an *incontestable* explanation to the applicant rather than reveal the true workings of the model since the explanation can be used by the applicant to prove discrimination in the court of law (Bordt et al., 2022). In fact, many previous studies show that adversarial manipulations of explanations are possible in realistic settings with systematic and computationally feasible attacks (Slack et al., 2020; Shahin Shamsabadi et al., 2022; Slack et al., 2021; Yadav et al., 2024c). As such, despite the demand, explanations fail to be operational as a trust-enhancing tool.

A major barrier to using explanations in adversarial contexts is that organizations keep their models confidential due to IP and legal reasons. However, confidentiality aids in manipulating explanations by allowing model swapping – a model owner can use different models for generating predictions vs. explanations, swap the model for specific inputs, or change the model post-audits (Slack et al., 2020; Yadav et al., 2024b; Yan & Zhang, 2022; Baldini et al., 2023). This problem demands a technical solution which guarantees that a specific model is used for all inputs, for generating both the prediction and the explanation and prove this to the customer on the receiving end while keeping the model weights confidential.

Another important barrier to using explanations in adversarial contexts is that many explanation algorithms are not

deterministic and have many tunable parameters. An adversary can choose these parameters adversarially to make discriminatory predictions seem benign. Moreover, there is no guarantee that the model developer is following the explanation algorithm correctly to generate explanations. A plausible solution to counter this problem is consistency checks (Bhattacharjee & von Luxburg, 2024; Dasgupta et al., 2022). Apart from being a rather lopsided ask where the onus of proving correctness of explanations lies completely on the customer, these checks require collecting multiple explanation-prediction pairs for different queries and are therefore infeasible for individual customers in the real world. Compounding the issue, it has been shown that auditing local explanations with consistency checks is hard (Bhattacharjee & von Luxburg, 2024). Note that many of these issues persist even with a perfectly faithful algorithm for generating explanations.

We address the aforementioned challenges by proposing a system called *ExpProof*. *ExpProof* gives a protocol consisting of (1) cryptographic commitments which guarantee that the same model is used for all inputs and (2) Zero-Knowledge Proofs (ZKPs) which guarantee that the explanation was computed correctly using a predefined explanation algorithm, both while maintaining model confidentiality. See Fig. 1 for a pictorial representation of *ExpProof*.

*ExpProof* ensures uniformity of the model and explanation parameters through cryptographic commitments (Blum, 1983). Commitments publicly bind the model owner to a fixed set of model weights and explanation parameters while keeping the model confidential. Commitments for ML models are a very popular and widely researched area in cryptography and hence we use standard procedures to do this (Kate et al., 2010).

Furthermore, we wish to provide a way to the customer to verify that the explanation was indeed computed correctly using the said explanation algorithm, without revealing model weights. To do this, we employ a cryptographic primitive called Succinct Zero-Knowledge Proofs (Goldwasser et al., 1985; Goldreich et al., 1991). ZKPs allow a prover (bank) to prove a statement (explanation) about its private data (model weights) to the verifier (customer) without leaking the private data. The prover outputs a cryptographic proof and the verifier on the other end verifies the proof in a computationally feasible way. In our case, if the proof passes the verifier's check, it means that the explanation was correctly computed using the explanation algorithm and commited weights without any manipulation.

How are explanations computed? The explanation algorithm we use in our paper is a popular one called LIME (Ribeiro et al., 2016), which returns a local explanation for the model decision boundary around an input point. We choose a local explanation rather than a global one since customers are often more interested in the behavior of the model around their input specifically. Additionally, LIME is model-agnostic, meaning that it can be used for any kind of model class (including non-linear ones).

Traditionally ZKPs are slow and are infamous for adding a huge computational overhead for proving even seemingly simple algorithmic steps. Moreover many local explanation algorithms such as LIME require solving an optimization problem and involve non-linear functions such as exponentials, which makes it infeasible to simply reimplement LIME as-is in a ZKP library. To remedy these issues, we experiment with different variants of LIME exploring the resulting tradeoffs between explanation-fidelity and ZKP-overhead. To make our ZKP system efficient, we also utilize the fact that verification can be easier than re-running the computation – instead of *solving* the optimization problem within ZKP, we verify the optimal solution using duality gap.

**Experiments.** We evaluate *ExpProof* on fully connected ReLU Neural Networks and Random Forests for three standard datasets on an Ubuntu Server with 64 CPUs of x86_64 architecture and 256 GB of memory without any explicit parallelization. Our results show that *ExpProof* is computationally feasible, with a maximum proof generation time of 1.5 minutes, verification time of 0.12 seconds and proof size of 13KB for NNs and standard LIME.

## 2. Preliminaries

**Cryptographic Primitives.** We use two cryptographic primitives in this paper, commitment schemes and Zero-Knowledge Proofs.

Commitment Scheme (Blum, 1983) commits to private inputs $w$ outputting a commitment string $\mathsf{com}_w$. A commitment scheme is *hiding* meaning that $\mathsf{com}_w$ does not reveal anything about private input $w$ and *binding* meaning that there cannot exist another input $w'$ which has the commitment $\mathsf{com}_w$, binding commitment $\mathsf{com}_w$ to input $w$.

Zero-Knowledge Proofs (ZKPs) (Goldwasser et al., 1985; Goldreich et al., 1991) involve a prover holding a private input $w$, and a verifier who both have access to a circuit $P$. ZKPs enable the prover to convince the verifier that, for some public input $x$, it holds a private witness $w$ such that $P(x, w) = 1$ without revealing any additional information about witness $w$ to the verifier. A ZKP protocol is (1) *complete*, meaning that for any inputs $(x, w)$ where $P(x, w) = 1$, an honest prover will always be able to convince an honest verifier that $P(x, w) = 1$ by correctly following the protocol, (2) *sound*, meaning that beyond a negligible probability, a malicious prover cannot convince an honest verifier for any input $x$, that for some witness $w$, $P(x, w) = 1$ when infact such a witness $w$ does not exist, even by arbitrarily deviating from the protocol, and

(3) *zero-knowledge*, meaning that for any input $x$ and witness $w$ such that $P(x, w) = 1$, a malicious verifier cannot learn any additional information about witness $w$ except that $P(x, w) = 1$ even when arbitrarily deviating from the protocol. A classic result says that any predicate P in the class NP can be verified using ZKPs (Goldreich et al., 1991).

**LIME.** Existing literature has put forward a wide variety of post-hoc (post-training) explainability techniques to make ML models transparent. In this paper, we focus on one of the popular ones, LIME (stands for Local Interpretable Model-Agnostic Explanations) (Ribeiro et al., 2016).

LIME explains the prediction for an input point by approximating the local decision boundary around that point with a linear model. Formally, given an input point $x \in \mathcal{X}$, a complex non-interpretable classifier $f : \mathcal{X} \to \mathcal{Y}$ and an interpretable class of models $\mathcal{G}$, LIME explains the prediction $f(x) \in \mathcal{Y}$ with a local interpretable model $g \in \mathcal{G}$. The interpretable model $g$ is found from the class $\mathcal{G}$ via learning, on a set of points randomly sampled around the input point and weighed according to their distance to the input point, as measured with a similarity kernel $\pi$. The similarity kernel creates a locality around the input by giving higher weights to samples near input $x$ as compared to those far off. A natural and popular choice for the interpretable class of models $\mathcal{G}$ is linear models such that for any $g \in \mathcal{G}$, $g(z) = w_g \cdot z$ ((Ribeiro et al., 2016; Garreau & von Luxburg, 2020)), where $w_g$ are the coefficients of linear model $g$. These coefficients highlight the contribution of each feature towards the prediction and therefore serve as the explanation in LIME. Learning the linear model is formulated as a weighted LASSO problem, since the sparsity induced by $\ell_1$ regularization leads to more interpretable and human-understandable explanations. Following (Ribeiro et al., 2016), the similarity kernel is set to be the exponential kernel with $\ell_2$ norm as the distance function, $\pi_x(z) = \exp\left(-\ell_2(x, z)^2/\sigma^2\right)$ where $\sigma$ is the bandwidth parameter of the kernel and controls the locality around input $x$.

For brevity, we will denote the coefficients corresponding to the linear model $g$ as $w$ instead of $w_g$, unless otherwise noted. For readers familiar with LIME, without loss of generality we consider the transformation of the points into an interpretable feature space to be identity in this paper for simplicity of exposition. The complete LIME algorithm with linear explanations is given in Alg. 1. We will also use 'explanations' to mean post-hoc explanations throughout the rest of the paper.

## 3. Problem Setting & Desiderata for Solution

To recall, explanations fail as a trust-enhancing tool in adversarial use-cases and can lead to a false sense of security

---

**Algorithm 1** LIME (Ribeiro et al., 2016)
1: **Input:** Input point $x$, Classifier $f$
2: **Parameters:** Number of points $n$ to be sampled around input point, Length of explanation $K$, Bandwidth parameter $\sigma$ for similarity kernel
3: **Output:** Explanation $e$
4:
5: **for** $i \in \{1, 2, 3, \ldots, n\}$ **do**
6:     $z_i \leftarrow \text{sample\_around}(x)$
7:     $\pi_i \leftarrow \exp\left(-\ell_2(x, z_i)^2/\sigma^2\right)$
8: **end for**
9: $\hat{w} \in \arg\min_w \sum_{i=1}^{n} \pi_i \times \left(f(z_i) - w^\top z_i\right)^2 + \|w\|_1$
10: $e := \text{top-K}(\hat{w}, K)$       ▷ Sorts the weights according to absolute value & returns these along with corresponding features
11: Return Explanation $e$

---

while benefiting adversaries. Motivated by these problems, we investigate if a technical solution can be designed to operationalize explanations in adversarial settings.

**Formal Problem Setting.** Formally, a model owner confidentially holds a classification model $f$ which is not publicly released due to legal and IP reasons. A customer supplies an input $x$ to the model owner, who responds with a prediction $f(x)$ and an explanation $\mathcal{E}(f, x)$ where $\mathcal{E}$ is the possibly-randomized algorithm generating the explanation.

**Threat Model.** The model developer having access to the training data trains the model honestly. The parameters of LIME are public and are assumed to be set honestly. Model developer doesn't have information on the input queries it will receive. When presented with an input query, the model generates both the prediction and explanation, which could be manipulated by either changing the model prediction arbitrarily, using a different model than the one which trained or using a different/corrupted algorithm to generate the explanation.

**Solution Desiderata.** A technical solution to operationalize explanations in adversarial use-cases should provide the following guarantees.

1. (Model Uniformity) the same model $f$ is used by the model owner for all inputs : our solution is to use cryptographic commitments which force the model owner to commit to a model prior to receiving inputs,

2. (Explanation Correctness) the explanation algorithm $\mathcal{E}$ is run correctly for generating explanations for all inputs : our solution is to use Zero-Knowledge Proofs, wherein the model owner supplies a cryptographic proof of correctness to be verified by the customer in a computationally feasible manner,

**Algorithm 2** STANDARD_LIME_VARIANTS
1: **Input:** Input point $x$, Classifier $f$
2: **Parameters:** Number of points $n$ to be sampled around input point, Length of explanation $K$, Bandwidth parameter $\sigma$ for similarity kernel, sampling type $smpl\_type$, kernel type $krnl\_type$
3: **Output:** Explanation $e$
4:
5: **for** $i \in \{1, 2, 3, \ldots, n\}$ **do**
6:    **if** $smpl\_type$=='uniform' **then**
7:       $z_i \leftarrow$ uniformly_sample_around$(x)$
8:    **else if** $smpl\_type$=='gaussian' **then**
9:       $z_i \leftarrow$ gaussian_sample_around$(x)$
10:    **end if**
11:    **if** $krnl\_type$=='exponential' **then**
12:       $\pi_i \leftarrow \exp\left(-\ell_2(x, z_i)^2/\sigma^2\right)$
13:    **else**
14:       $\pi_i = 1$
15:    **end if**
16: **end for**
17: $\hat{w} \in \arg\min_w \sum_{i=1}^n \pi_i \times \left(f(z_i) - w^\top z_i\right)^2 + \|w\|_1$
18: $e := \text{top-K}(\hat{w}, K)$
19: Return Explanation $e$

---

3. (Model Consistency) the same model $f$ is used for inference and generating explanations : this is ensured by generating inference and explanations as a part of the same system and by using model commitments,

4. (Model Weights Confidentiality) the model weights of $f$ are kept confidential in the sense that any technique for guaranteeing (1)-(3) does not leak anything else about the hidden model weights than is already leaked by predictions $f(x)$ and explanations $\mathcal{E}(f, x)$ without using the technique : this comes as a by-product of using ZKPs and commitments (See Sec. A.1 for the formal theorem and proof),

5. (Technique Reliability) the technique used for guaranteeing (1)-(4) is sound and complete (as in Sec.2): this comes as a by-product of using ZKPs and commitments (See Sec. A.1 for the formal theorem and proof).

Our solution *ExpProof* which provides the above guarantees will be discussed in Sec. 5.

## 4. Variants of LIME

Building zero-knowledge proofs of explanations requires the explanation algorithm to be implemented in a ZKP library[1]

---

[1]More precisely, arithmetic circuits for the explanation algorithm are implemented in the ZKP library.

which is known to introduce a significant computational overhead. Given this, a natural question that comes to mind is if there exist variants of LIME which provide similar quality of explanations but are more ZKP-amenable by design, meaning they introduce a smaller ZKP overhead?

**Standard LIME Variants.** To create variants of standard LIME (Alg.1), we focus on the two steps which are carried out numerous times and hence create a computational bottleneck in the LIME algorithm – sampling around input $x$ (Step 6 in Alg. 1) and computing distance using exponential kernel (Step 7 in Alg. 1). For sampling, we propose two options as found in the literature : gaussian (G) and uniform (U) (Ribeiro et al., 2016; Garreau & Luxburg, 2020; Garreau & von Luxburg, 2020). For the kernel we propose to either use the exponential (E) kernel or no (N) kernel. These choices give rise to four variants of LIME, mentioned in Alg. 2. We address each variant by the intials in the brackets, for instance standard LIME with uniform sampling and no kernel is addressed as 'LIME_U+N'.

**BorderLIME.** An important consideration for generating meaningful local explanations is that the sampled neighborhood should contain points from different classes (Laugel et al., 2018b). Any reasonable neighborhood for an input far off from the decision boundary will only contain samples from the same class, resulting in vacuous explanations.

To remedy the problem, (Laugel et al., 2018a;b) propose a *radial* search algorithm, which finds the closest point to the input $x$ belonging to a different class, $x_{border}$, and then uses $x_{border}$ as the input to LIME (instead of original input $x$). Their algorithm incrementally grows (or shrinks) a search area radially from the input point and relies on random sampling within each 'ring' (or sphere), looking for points with an opposite label. To cryptographically prove this algorithm, we would either have to reimplement the algorithm as-is or would have to give a probabilistic security guarantee (using a concentration inequality), both of which would require many classifier calls and thereby many proofs of inference, becoming inefficient in a ZKP system.

We transform their algorithm into a line search version, called BorderLIME, given in Alg. 3 and 4, using the notion of Stability Radius which is now fed as a parameter to the algorithm. The stability radius for an input $x$, $\delta_x$, is defined as the largest radius for which the model prediction remains unchanged within a ball of that radius around the input $x$. The stability radius $\delta$ is defined as the minimum stability radius across all inputs $x$ sampled from the data distribution $D$. Formally, $\delta = \inf_{x \sim \mathcal{D}} \delta_x$, where $\delta_x = \sup\{r \geq 0 \mid f(x') = f(x), \forall x' \in \mathcal{B}(x, r)\}$. Here $\mathcal{B}(x, r) = \{x' \mid \|x' - x\| \leq r\}$ denotes a ball of radius $r$ centered at $x$. Stability radius ensures that for any input from the data distribution, the model's prediction remains stable within at least a radius of $\delta$.

Our algorithm samples $m$ directions and then starting from the original input $x$, takes $\delta$ steps until it finds a point with a different label along all these directions individually. The border point $x_{border}$ is that oppositely labeled point which is closest to the input $x$. Furthermore, unlike in the algorithm in (Laugel et al., 2018a), our algorithm can exploit parallelization by searching along the different directions in parallel since these are independent.

Determining the optimal value of the stability radius is an interesting research question, but it is not the focus of this work. We leave an in-depth exploration of this topic to future work while providing some high-level directions and suggestions next. Stability radius can (and perhaps should) be found *offline* using techniques as proposed in (Yadav et al., 2024a; Jordan et al., 2019) or through an offline empirical evaluation on in-distribution points. A ZK proof for this radius can be generated one-time, in an offline manner and supplied by the model developer (for NNs see (Yadav et al., 2024a)). It can also be pre-committed to by the model developer (see Sec. 5).

---

**Algorithm 3** BORDERLIME

1: **Input:** Input point $x$, Classifier $f$
2: **Parameters:** Number of points $n$ to be sampled around input point, Length of explanation $K$, Bandwidth parameter $\sigma$ for similarity kernel
3: **Output:** Explanation $e$
4:
5: $x_{border} :=$
        FIND_CLOSEST_POINT_WITH_OPP_LABEL$(x, f)$
        $\triangleright$ See Alg. 4
6: $e := \text{LIME}(x_{border}, f)$ $\triangleright$ Note that any variant of LIME can be used here.
7: Return Explanation $e$

---

## 5. *ExpProof*: Verification of Explanations

Our system for operationalizing explanations in adversarial settings, *ExpProof*, consists of two phases: (1) a One-time Commitment phase and (2) an Online verification phase which should be executed for every input.

**Commitment Phase.** To ensure model uniformity, the model owner cryptographically commits to a fixed set of model weights $\mathbf{W}$ belonging to the original model $f$, resulting in committed weights $\mathsf{com_W}$. Architecture of model $f$ is assumed to be public. Additionally, model owner can also commit to the values of different parameters used in the explanation algorithm or these parameters can be public.

**Online Verification Phase.** This phase is executed every time a customer inputs a query. On receiving the query, the prover (bank) outputs a prediction, an explanation and a zero-knowledge proof of the explanation. Verifier (cus-

---

**Algorithm 4** FIND_CLOSEST_POINT_WITH_OPP_LABEL

1: **Input:** Input point $x$, Classifier $f$
2: **Parameters:** Number of random directions $m$, Stability radius $\delta$, Iteration Threshold $T$
3: **Output:** Opposite label point $x_{border}$
4:
5: $\{\vec{u}_0, \vec{u}_1 \cdots \vec{u}_{m-1}\} :=$ Sample $m$ random directions
6: Initialize $dist_0 \cdots dist_{m-1}$ as inf
7: **for** $\vec{u}_i \in \{\vec{u}_0, \vec{u}_1 \cdots \vec{u}_{m-1}\}$ **do**
8:     $x_{border_i} := x$
9:     $iter := 0$
10:    **while** $f(x_{border_i}) == f(x)$ and $iter \leq T$ **do**
11:        $x_{border_i} := x_{border_i} + \delta\vec{u}_i$
12:        $iter := iter + 1$
13:    **end while**
14:    **if** $f(x_{border_i})! = f(x)$ **then**
15:        $dist_i := \ell_2(x, x_{border_i})$
16:    **end if**
17: **end for**
18: $x_{border} := x_{border_i}$ such that $i := \arg\min dist_i$
19: Return $x_{border}$

---

tomer) validates the proof without looking at the model weights. If the proof passes verification, it means that the explanation is correctly computed with the committed model weights and explanation algorithm parameters.

To generate the explanation proof, a ZKP circuit which implements (a variant of) LIME is required. However since ZKPs can be computationally inefficient, instead of reimplementing the algorithm as-is in a ZKP library, we devise some smart strategies for verification, based on the fact that verification can be easier than redoing the computation. Since all the variants of LIME share some common functionalities, we next describe how the verification strategies for these functionalities. For more details on the verification for each variant, see Appendix Sec. A.

*1. Verifying Sampling (Alg. 7, 12, 13).* We use the Poseidon (Grassi et al., 2021) hash function to generate random samples. As part of the setup phase, the prover commits to a random value $r_p$. When submitting an input for explanation, the verifier sends another random value $r_v$. Prover generates uniformly sampled points using Poseidon with a key $r_p + r_v$, which is uniformly random in the view of both the prover and the verifier. Then, during the proof generation phase, the prover proves that the sampled points are the correct outputs from Poseidon using *ezkl*'s inbuilt efficient Poseidon verification circuit. We convert the uniform samples into Gaussian samples using the inverse CDF, which is checked in the proof using a look-up table for the inverse CDF.

*2. Verifying Exponential Kernel (Alg. 11).* ZKP libraries do not support many non-linear functions such as exponential,

which is used for the similarity kernel in LIME (Step 5 of Alg.1). To resolve this problem, we implement a look-up table for the exponential function and prove that the exponential value is correct by comparing it with the value from the look-up table.

*3. Verifying Inference.* Since LIME requires predictions for the sampled points in order to learn the linear explanation, we must verify that the predictions are correct. To generate proofs for correct predictions, we use *ezkl*'s inbuilt inference verification circuit.

*4. Verifying LASSO Solution (Alg. 9).* ZKP libraries only accept integers and hence all floating points have to be quantized. Consequently, the LASSO solution for Step 7 of Alg. 1 is also quantized in a ZKP library, leading to small scale differences between the exact and quantized solutions. To verify optimality of the quantized LASSO solution, we use the standard concept of duality gap. For a primal objective $l$ and its dual objective $g$, to prove that the objective value from primal feasible $w$ is close to that from the primal optimal $w^*$, that is $l(w) - l(w^*) \leq \epsilon$, the duality gap should be smaller than $\epsilon$ as well, $l(w) - g(u, v) \leq \epsilon$ where $u, v$ are dual feasible. Since the primal and dual of LASSO have closed forms, as long we input any dual feasible values, we can verify that the quantized LASSO solution is close to the LASSO optimal. The prover provides the dual feasible as part of the witness to the proof. See App. Sec.A.2 for a note on the meaning of the duality gap threshold $\epsilon$, how it should be chosen, closed forms of the primal and dual functions and for the technique to find dual feasible.

The complete *ExpProof* protocol can be found in Alg. 5; its security guarantee is given as follows.

**Theorem 5.1.** *(Informal) Given a model $f$ and an input point $x$, ExpProof returns prediction $f(x)$, LIME explanation $\mathcal{E}(f, x)$ and a ZK proof for the correct computation of the explanation, without leaking anything additional about the weights of model $f$ (in the sense described in Sec.3).*

For the complete formal security theorem and proof, refer to App. Sec. A.1.

## 6. Experiments

**Datasets & Models.** We use three standard fairness benchmarks for experimentation : Adult (Becker & Kohavi, 1996), Credit (Yeh, 2016) and German Credit (Hofmann, 1994). Adult has 14 input features, Credit has 23 input features, and German has 20 input features. All the continuous features in the datasets are standardized. We train two kinds of models on these datasets, neural networks and random forests. Our neural networks are 2-layer fully connected ReLU activated networks with 16 hidden units in each layer, trained using Stochastic Gradient Descent in PyTorch (Paszke et al., 2019)

with a learning rate of 0.001 for 400 epochs. The weights and biases are converted to fixed-point representation with four decimal places for making them compatible with ZKP libraries which do not work with floating points, leading to a $\sim 1\%$ test accuracy drop. Our random forests are trained using Scikit-Learn (Pedregosa et al., 2011) with 5-6 decision trees in each forest.

**ZKP Configuration.** We code *ExpProof* with different variants of LIME in the *ezkl* library (Konduit, 2024) (Version 18.1.1) which uses Halo2 (Zcash Foundation, 2023) as its underlying proof system in the Rust programming language, resulting in $\sim 3.7k$ lines of code. Our ZKP experiments are run on an Ubuntu server with 64 CPUs of x86_64 architecture and 256 GB of memory, without any explicit parallelization –while *ezkl* automatically does multithreading on all the available cores, we do not use GPUs, do not modify *ezkl* to do more parallelization and do not do any of the steps in ZK_LIME (Alg. 6) in parallel by ourselves. We use default configuration for *ezkl*, except for 200k rows for all lookup arguments in *ezkl* and *ExpProof*. We use KZG (Kate et al., 2010) commitments for our scheme that are built into *ezkl*.

**Research Questions & Metrics.** We ask the following questions for the different variants of LIME.

Q1) How faithful are the explanations generated by the LIME variant?
Q2) What is the time and memory overhead introduced by implementing the LIME variant in a ZKP library?

To answer Q1, we need a measure of fidelity of the explanation, we use 'Prediction Similarity' defined as the similarity of predictions between the explanation classifier and the original model in a local region around the input. We first sample points from a local[2] region around the input point, then classify these according to both the explanation classifier and the original model and report the fraction of matches between the two kinds of predictions as prediction similarity. In our experiments, the local region is created by sampling 1000 points from a Uniform distribution of half-edge length 0.2 or a Gaussian distribution centered at the input point with a standard deviation of 0.2.

To answer Q2, we will look at the proof generation time taken by the prover to generate the ZK proof, the verification time taken by the verifier to verify the proof and the proof length which measures the size of the generated proof.

### 6.1. Standard LIME Variants

In this section we compare the different variants of Standard LIME, given in Alg. 2 Sec. 4, w.r.t. the fidelity of their

---

[2]Note that this local region is for evaluation and is different from the local neighborhood in LIME.

explanations and ZKP overhead.

*Setup.* We use the LIME library for experimentation and run the different variants of LIME with number of neighboring samples $n = 300$ and length of explanation $K = 5$. Based on the sampling type, we either sample randomly from a hypercube with half-edge length as 0.2 or from a gaussian distribution centered around the input point with a standard deviation of 0.2. Based on the kernel type we either do not use a kernel or use the exponential kernel with a bandwidth parameter as $\sqrt{\#features} * 0.75$ (default value in the LIME library). Rest of the parameters also keep the default values of the LIME library. Our results are averaged over 50 different input points sampled randomly from the test set. The duality gap constant, $\epsilon$ is set to 0.001.

Results for NNs with 300 neighboring samples and Gaussian sampling for fidelity evaluation are described below. Results for uniform sampling fidelity evaluation, fidelity evaluation with neighborhood $n = 5000$ points and all results for RFs can be found in the Appendix Sec. B.

*Fidelity Results.* As shown in Fig. 2 left, we do not find a huge difference between the explanation fidelities of the different variants of LIME as the error bars significantly overlap. This could be due to the small size of the local neighborhoods where the kernel or sampling doesn't matter much. However, for the credit dataset, which has the highest number of input features, gaussian sampling works slightly better than uniform, which could be because of the worsening of uniform sampling with increasing dimension.

*ZKP Overhead Results.* Across the board, proof generation takes a maximum of $\sim 1.5$ minutes, verification time takes a maximum of $\sim 0.12$ seconds and proof size is a maximum of $\sim 13$KB, as shown in Fig. 3. Note that while proof generation time is on the order of minutes, verification time is on the order of seconds – this is due to the inherent design of ZKPs, requiring much lesser resources at the verifier's end (contrary to consistency-based explanation checks). We also observe that the dataset type does not have much influence on the ZKP overhead; this is due to same ZKP backend parameters needed across datasets.

Furthermore, we see that gaussian sampling leads to a larger ZKP overhead. This can be attributed to our implementation of gaussian sampling in the ZKP library, wherein we first create uniform samples and then transform them to gaussian samples using the inverse CDF method, leading to an additional step in the gaussian sampling ZKP circuit as compared to that of uniform sampling. Similarly, using the exponential kernel leads to a larger overhead over not using it due to additional steps related to verifying the kernel.

Overall, 'gaussian sampling and no kernel' variant of LIME is likely the most amenable for a practical ZKP system as it produces faithful explanations with a small overhead.

## 6.2. BorderLIME

In this section we compare the variants of BorderLIME (Alg. 3, Sec. 4) and BorderLIME vs. Standard LIME w.r.t. the fidelity of their explanations and ZKP overheads.

*Setup.* We implement BorderLIME with all of the Standard LIME variants (Step 6 of Alg. 3). For the purpose of experimentation, we fix the iteration threshold to $T = 250$, number of directions to $m = 5$. Then to *approximate* the stability radius $\delta$, we incrementally go over the set $\{0.01, 0.03, 0.05, 0.07, 0.1, 0.15\}$ and use the smallest value for which an opposite class point is found for all 50 randomly sampled input points. While this is a heuristic approach and does not guarantee the theoretically minimal stability radius, it provides a practical estimate of stability radius for efficient experimentation. Once a suitable value of stability radius is identified, we tighten the number of directions by reducing them while ensuring that at least one opposite-class point exists for each input. Our results are averaged over 50 input points. The exact parameter values used in our final setup can be found in App. Sec. B.

Results for NNs with 300 neighboring samples and Gaussian sampling for fidelity evaluation are described below. Results for uniform sampling fidelity evaluation, fidelity evaluation with neighborhood $n = 5000$ points and all results for RFs can be found in the Appendix Sec. B.

*Fidelity Results.* Comparing different variants of Border-LIME based on the LIME implementation, we observe that the difference in explanation fidelity between gaussian and uniform sampling becomes more pronounced compared to standard LIME as shown in Fig. 2, reinforcing the importance of gaussian sampling. This gap can sometimes be reduced by using more neighborhood points, i.e. a larger $n$, when uniformly sampling. As demonstrated in Fig. 2 mid, with three times more points for uniform sampling, we can match the fidelity of gaussian explanations for Adult and German datasets. Comparing the G+N version of Border-LIME and standard LIME in Fig. 2 right, we observe that explanations generated by BorderLIME are atleast as faithful as standard LIME and can sometimes be better hinting to its capability of generating more meaningful explanations.

*ZKP Overhead Results.* We observe that BorderLIME has a larger ZKP overhead than standard LIME as shown in Table 2; this can be attributed to the additional steps needed in BorderLIME to find the border point with opposite label (Alg. 4) which also have to be proved and verified. Similar to the previous subsection, the overhead is similar across datasets and verification is orders of magnitude cheaper than proof generation.

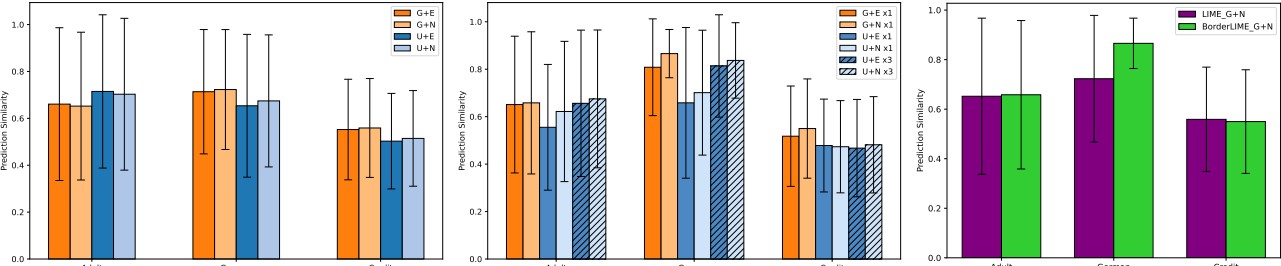

*Figure 2.* Results for NNs. G/U: gaussian or uniform sampling, E/N: using or not using the exponential kernel. Left: Fidelity of different variants of Standard LIME, Mid: Fidelity of different variants of BorderLIME , Right: Fidelity of Standard vs. BorderLIME.

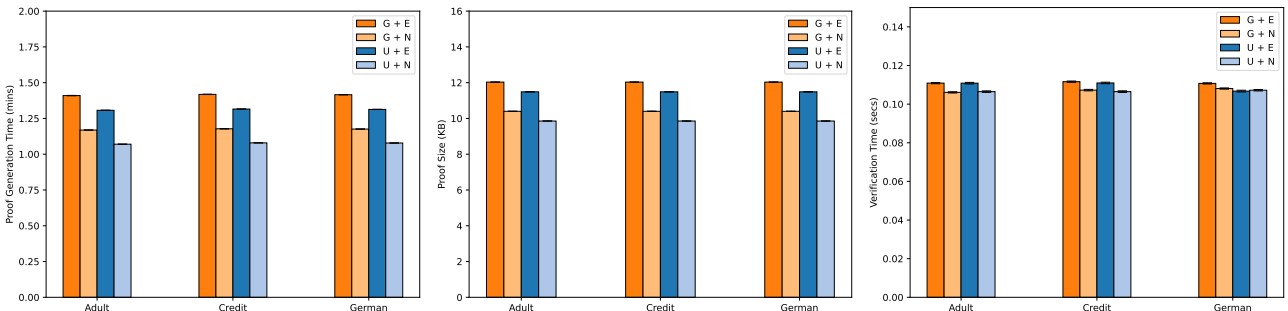

*Figure 3.* Results for NNs. G/U: gaussian or uniform sampling, E/N: using or not using the exponential kernel. Left: Proof Generation Time (in mins), Mid: Proof Size (in KBs), Right: Verification times (in secs) for different variants of Standard LIME. All configurations use the same number of Halo2 rows, $2^{18}$, and lookup tables of size 200k.

## 6.3. Ablation Study

We conduct an ablation study to investigate the computational bottleneck of LIME when implemented in a ZKP library, ZK_LIME Alg. 6, and understand how the computational overhead of ZK_LIME scales with model size.

**Setup.** We present experiments with the Credit dataset (w.l.o.g.) since this dataset has the highest feature dimensionality and for NN models. For finding the computational bottleneck of ZK_LIME, we fix the number of hidden units in each layer of the NN to 16 while changing the number of layers to $\{2, 10, 20, 40\}$. For studying the scaling, we change the number of units in each hidden layer to $\{16, 32, 64\}$ and the number of hidden layers to $\{2, 5, 10, 20\}$. We use the G+E variant of LIME (as it is the most computationally expensive variant, from previous experiments) with a sampling neighborhood of $n = 300$. Results are displayed for one input point picked randomly as the variance in ZKP cost is almost zero, as seen from previous experiments.

**Computational Bottleneck of ZK_LIME.** To understand which part of ZK_LIME is the most expensive, we split it into three broad categories : (1) Sampling and applying exponential kernel (steps 13 & 26 Alg. 6), (2) Inference

(steps 9 & 30-32 Alg. 6) and (3) LASSO (step 33 Alg. 6). For these three, we provide the Proof-Generation time in Table 1, since proof generation is the most time-consuming part of a ZKP system and the trends for verification time and proof sizes usually follow its trend.

As demonstrated in Table 1, while the proof-generation time for Sampling+Kernel and LASSO is almost same across different model sizes, the time for Inference proofs increases polynomially and dominates the total proof-generation time with increasing number of hidden layers. Since we use inference proof engines as a plug-and-play module in ZK_LIME, the advancements in inference proof generation (which is a very active research area) will directly translate to ZK_LIME. Current SOTA inference proof generation time is ~15 minutes for a 13B model (Sun et al., 2024).

**ZKP-Overhead Scaling with Model Size.** As demonstrated in Fig.4, the total proof-generation and verification times increase polynomially with increasing number of hidden layers and units in each hidden layer. Feature Dimensionality has a less pronounced effect as demonstrated in Fig.3 where the datasets have different dimensions (as the internal ZKP parameters (rows, columns) change when the dimension exceeds a threshold).

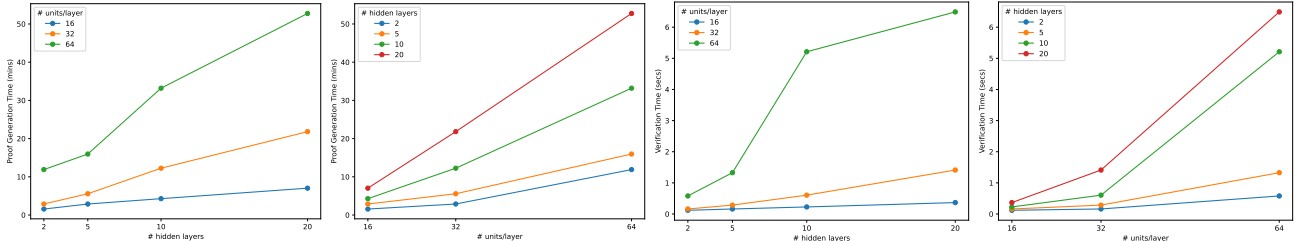

*Figure 4.* Proof-Generation (mins) and Verification Time (secs) with increasing #HiddenLayers and #Units/HiddenLayer.

| #Layers | Sampling+ExpKernel | LASSO | Inferences |
|---------|--------------------|-------|------------|
| 2 | 23.54 | 59.62 | 42.15 |
| 10 | 24.11 | 60.34 | 108.23 |
| 20 | 24.50 | 60.70 | 179.00 |
| 40 | 25.59 | 62.30 | 319.48 |

*Table 1.* Proof-Generation Time Split (in seconds) for ZK_LIME. Inference time increases with #HiddenLayers while Sampling+Kernel and LASSO times remain constant.

| ZKP Overhead Type | BorderLIME | LIME |
|-------------------|-----------|------|
| Proof Generation Time (mins) | $4.85 \pm 10^{-2}$ | $1.17 \pm 10^{-2}$ |
| Verification Time (secs) | $0.30 \pm 10^{-2}$ | $0.11 \pm 10^{-2}$ |
| Proof Size (KB) | $18.30 \pm 0$ | $10.40 \pm 0$ |

*Table 2.* ZKP Overhead of BorderLIME and Standard LIME (both G+N variant) for NNs. Overhead for BorderLIME is larger than that for LIME. Results are consistent across all datasets.

## 7. Discussion

While *ExpProof* guarantees model and parameter uniformity as well as correctness of explanations for a given model, it cannot prevent the kinds of manipulation where the model itself is corrupted – the model can be trained to create innocuous explanations while giving biased predictions. Here usually a regularization term corresponding to the manipulated explanations is added to the loss function (Aïvodji et al., 2019; Yadav et al., 2024c). Preventing such attacks requires a ZK proof of training; this is well-studied in the literature but is outside the scope of this work and we refer an interested reader to (Garg et al., 2023).

Furthermore, to provide end-to-end trust guarantees for fully secure explanations, the explanations should be (1) faithful, stable and reliable, (2) robust to realistic adversarial attacks (such as the one mentioned above) and (3) should also be verifiable under confidentiality. This paper looks at the third condition by giving a protocol *ExpProof* and implementing it for verifiable explanations under confidentiality, which has not been studied prior to our work. As such, we view *our work as complementary and necessary* for end-to-end

explanation trust guarantees.

## 8. Related Work

In the ML field ZKPs have been majorly used for verification of inferences made by models (Sun et al., 2024; Chen et al., 2024; Kang et al., 2022; Weng et al., 2023; Sun & Zhang, 2023; Feng et al., 2021; VI2, 2023; Lee et al., 2020; Zhang et al., 2020a; Liu et al., 2021; Singh et al., 2022; Fan et al., 2023). A line of work also focuses on proving the training of ML models using ZKPs (Lycklama et al., 2023; Huang et al., 2022; Rückel et al., 2022; Garg et al., 2023; Abbaszadeh et al., 2024). More recently they're also been used for verifying properties such as fairness (Yadav et al., 2024a; Shamsabadi et al., 2023; Toreini et al., 2023) and accuracy (Zhang et al., 2020b) of confidential ML models. Alternatively, (Waiwitlikhit et al., 2024) propose ZKPs as a general technique for auditing ML models broadly. Contrary to these and to the best of our knowledge, ours is the first work that identifies the need for proving explanations and provides ZKP based solutions for the same.

## 9. Conclusion & Future Work

In this paper we take a step towards operationalizing explanations in adversarial contexts where the involved parties have misaligned interests. We propose a protocol *ExpProof* using Commitments and Zero-Knowledge Proofs, which provides guarantees on the model used and correctness of explanations in the face of confidentiality requirements. We propose ZKP-efficient versions of the popular explainability algorithm LIME and demonstrate the feasibility of *ExpProof* for Neural Networks & Random Forests.

An interesting avenue for future work is the tailored design of explainability algorithms for high ZKP-efficiency and inherent robustness to adversarial manipulations. Another interesting avenue is finding other applications in ML where ZKPs can ensure verifiable computation and provide trust guarantees without revealing sensitive information.

## Impact Statement

This work takes a step towards operationalizing explanations in adversarial settings where the model is kept confidential from customers. With this work it can be guaranteed that the said committed model (1) is used for all inputs, (2) is used for generating predictions and explanations (3) cannot be swapped post-audits. It can also be guaranteed that the explanation is generated correctly using the said explanation algorithm. All of this is guaranteed while maintaining model confidentiality.

While our protocol *ExpProof* provides the above guarantees, in order to have complete trust guarantees we also require stable and faithful explanation algorithms which are robust to realistic adversarial attacks as mentioned in Sec.7. We mention existing solutions to tackle some of these issues in Sec.7 and Sec.4 and call for more research in these directions. These research directions though interesting and important are out of scope for our work.

## Acknowledgements

CY and KC would like to thank National Science Foundation NSF (CIF-2402817, CNS-1804829), SaTC-2241100, CCF-2217058, ARO-MURI (W911NF2110317), and ONR under N00014-24-1-2304 for research support. DB and EL were partially supported by NSF, DARPA, and the Simons Foundation. Opinions, findings, and conclusions or recommendations expressed in this material are those of the authors and do not necessarily reflect the views of DARPA.

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

# A. *ExpProof*

---

**Algorithm 5** *ExpProof*: Provable Explanation for Confidential Models

---

1: **Public Configuration:** ZK_LIME configuration $cc = ($ $smpl\_type$ sampling type, LIME kernel variant $krnl\_type$, whether to use border LIME $border\_lime$, standard deviation $\sigma$, model architecture $f$, LIME $\ell_1$ penalty $\alpha$, maximum dual gap $\epsilon$ )

2: **Public Input:** Input point $x$

3: **Private Witness:** Model weights $\mathbf{W}$

4: **Output:** Output label $o$, Explanation $e$, Proof $\pi$ that $o = f(x)$ and that $e$ is a valid LIME explanation.

5: **Pre-Processing Offline Phase**

6: Sample randomness $r \leftarrow \mathbb{F}$

7: Commit to the randomness $\mathsf{com_r}$ and release it publicly

8: Commit to the model weights $\mathsf{com_W}$ and release it publicly

9: **Online Phase**

10: $o = f(\mathbf{W}, x)$

11: Compute $h_i = \text{Poseidon}(k, i)$

12: Compute LIME perturbations $z$ from samples $s$

13: $y = f(\mathbf{W}, z)$

14: $(e, \hat{w}) \leftarrow \text{LIME}(f, y, z)$           $\triangleright$ Compute a LIME solution using perturbations $z$ with labels $y$

15: Compute a feasible dual solution $\hat{v}$ from $\hat{w}$

16: $\Pi \leftarrow \text{zkLime}(cc, x, o, r_v, C_r, C_\mathbf{W}, e; \mathbf{W}, y, h, \hat{w}, \hat{v})$

17: **return** $(o, e, \Pi)$

---

---

**Algorithm 6** ZK_LIME
___

1: **Public Configuration:** $smpl\_type$ sampling type, LIME kernel variant $krnl\_type$, whether to use border LIME $border\_lime$, standard deviation $\sigma$, model architecture $f$, LIME $\ell_1$ penalty $\alpha$, maximum dual gap $\epsilon$, sampling bit-width $b$

2: **Public Instance**: input point $x$, model output $o$, randomness $r_v$, commitment to the randomness $C_r$, commitment to the weights $C_\mathbf{W}$, $e$ top-k LIME features

3: **Private Witness**: Model weights $\mathbf{W}$, labels of the LIME samples $y$, hash outputs $h$, LIME model $\hat{w}$, LIME dual $\hat{v}$

4: **Output:** ZK Proof of the computation $\Pi$

5: Check that $C_r = \text{Com}(r_p)$

6: Check that $C_W = \text{Com}(\mathbf{W})$

7: $k \leftarrow r_p + r_v$

8: ZK_CHECK_POSEIDON$(h, k)$            $\triangleright$ Check that $h$ is generated from Poseidon using key $k$

9: EZKL.CHECK_INFERENCE$(f, \mathbf{W}, x, o)$            $\triangleright$ Check that $o = f(W, x)$ using EZKL

10: **if** $smpl\_type ==$ 'uniform' **then**

11:    $s \leftarrow$ ZK_UNIFORM_SAMPLE$(h, b)$        $\triangleright$ Check that $s$ is uniform generated from the hashes $h$

12: **else if** $smpl\_type ==$ 'gaussian' **then**

13:    $s \leftarrow$ ZK_GAUSSIAN_SAMPLE$(h, b)$      $\triangleright$ Check that $s$ is Gaussian generated from the hashes $h$

14: **else**

15:    return $\perp$

16: **end if**

17: **if** $border\_lime ==$ true **then**

18:    $x \leftarrow$ ZK_FIND_OPP_POINT$(x, s, num\_vectors, vector\_length)$

19:    $s \leftarrow s[m \times d...]$           $\triangleright$ Skip samples used for opposite point for fresh randomness

20: **end if**

21: **for** $i \in |z|$ **do**

22:    $j \leftarrow i \mod |x|$

23:    $z \leftarrow x_j + s_i - 2^{b-1}$                $\triangleright$ Perturb $x$ with samples $s$

24: **end for**

25: **if** $krnl\_type ==$ 'exponential' **then**

26:    $\pi \leftarrow$ ZK_EXPONENTIAL_KERNEL$(x, z, \sigma, \pi)$

27: **else**

28:    $\pi \leftarrow 1$

29: **end if**

30: **for** $i \in \{1, 2, 3, \ldots, n\}$ **do**

31:    EZKL.CHECK_INFERENCE$(f, \mathbf{W}, z_i, y_i)$        $\triangleright$ Check that $y_i = f(W, z_i)$ using EZKL

32: **end for**

33: ZK_LASSO$(z, y, \pi, \hat{w}, \hat{v}, \alpha)$

34: $e =$ ZK_TOP_K$(\hat{w})$

35: Generate proof $\Pi$ of the above computation (see note below).
___

By 'Generate proof $\Pi$ of the above computation', we mean that we eencode the above computation as a Halo2 relation, and use Halo2 to prove its correctness as a monolithic proof. Each sub-routine we call does not generate a separate proof, it is simply a sub-computation in the Halo2 relation, and we only split it up for organization purposes.

---

**Algorithm 7** ZK_CHECK_POSEIDON
___

1: **Input:** hashes $h$, key $k$

2: **Output:** True if each $h_i$ generated from Poseidon with key $k$ and input $i$

3: **for** $h_i \in h$ **do**

4:    EZKL.CHECK_POSEIDON$(h_i, k, i)$         $\triangleright$ Check that $h_i = \text{Poseidon}(k, i)$ using EZKL

5: **end for**

6: return $x_{border}$
___

---

**Algorithm 8** ZK_FIND_OPP_POINT

---

1: **Input:** input $x$, samples $s$, number of vectors $num\_vectors$, maximum length of each vector $vector\_length$
2: **Output:** Border point $x_{border}$ if one exists, otherwise $x$
3: $d \leftarrow |x|$
4: $step \leftarrow z[0..d \times m].reshape(m, d)$         $\triangleright$ Get $m \times d$ samples as $m$ randomly sampled vectors
5: **for** $i \in \{1, 2, 3, ..., num\_vectors\}$ **do**
6:     $step_i \leftarrow step_i \times$ LOOKUP_RECIPROCAL_SQRT$(step_i \cdot step_i)$$\triangleright$ Normalize each step vector using a lookup table for
     $1/\sqrt{\|(step_i)\|_2}$
7: **end for**
8: **for** $i \in \{1, 2, 3, ..., num\_vectors\}$ **do**
9:     **for** $i \in \{1, 2, ..., vector\_length$ **do**
10:        $v_i \leftarrow i \times step\_size \times step_i$
11:     **end for**
12: **end for**
13: $y = f(W, v)$
14: $x_{border} \leftarrow x$
15: **for** $i \in \{vector\_length, vector\_length - 1, ..., 2, 1\}$ **do**
16:     **for** $i \in \{1, 2, ..., num\_vectors\}$ **do**
17:        **if** $y_i \neq x\_label$ **then**
18:           $x_{border} \leftarrow v_i$
19:        **end if**
20:     **end for**
21: **end for**
22: return $x_{border}$

---

**Algorithm 9** ZK_LASSO

---

1: **Input:** Samples $z$, labels $y$, weights $\pi$, Lasso solution $\hat{w}$, Lasso dual solution $\hat{v}$, Lasso parameter $\alpha$, maximum dual gap
    $\epsilon$
2: **Output:** True if the dual solution is feasible and the dual gap is less than $\epsilon$, and False otherwise.
3: **for** $i \in \{1, 2, 3, ..., n\}$ **do**
4:     $z'_i \leftarrow \sqrt{\pi_i} \times z_i$
5:     $y'_i \leftarrow \sqrt{\pi_i} \times y_i$
6: **end for**
7: $p \leftarrow \frac{1}{2n}\|y' - b - w^T z'_i\|^2 + \alpha\|w\|_1$
8: $d \leftarrow \frac{-n}{2}\|v\|^2 + v^T(y' - b)$
9: Check $p - d \leq \epsilon$
10: Let $m$ be the length of each sample $z_i$
11: **for** $i \in \{1, 2, 3, ..., m\}$ **do**
12:     $f_i \leftarrow (X^T)_i v$
13:     Check $-\alpha \leq f_i \leq \alpha$
14: **end for**

---

**Algorithm 10** ZK_TOP_K

---

1: **Input:** Lasso solution $\hat{w}$, top-k values $e$
2: **Output:** True if $e$ contains the top-k values of $\hat{w}$, and False otherwise
3: $\hat{w}' = \text{Sort}(\hat{w})$
4: **for** $i \in \{1, 2, 3, ..., k\}$ **do**
5:     $(v, j) \leftarrow e_i$
6:     Check $\hat{w}'_i = v$
7:     Check $\hat{w}_j = v$
8: **end for**

---

---

**Algorithm 11** ZK_EXPONENTIAL_KERNEL

---

1: **Input:** Input point $x$, LIME samples $z$, standard deviation $\sigma$
2: **Output:**
3: **for** $i \in \{1, 2, 3, \ldots, N\}$ **do**
4:     square_distance $= x \cdot z_i$
5:     $\pi_i \leftarrow$ LOOKUP_EXPONENTIAL$(-$square_distance$/\sigma^2)$       ▷ Check exponential function using a lookup table
6: **end for**
7: return $\pi$

---

---

**Algorithm 12** ZK_UNIFORM_SAMPLE

---

1: **Input:** Poseidon hashes $h$, sampling bit-width $b$
2: **Output:** Uniform samples $z$
3: $B = \lfloor W/b \rfloor$
4: $N = \lceil (|x| * n)/B \rceil$
5: $j = 0$
6: **for** $i \in \{1, 2, 3, \ldots, N\}$ **do**
7:     Compute $z$ such that $z_j + z_{j+1}2^b + \ldots + z_{j+B}2^{B*b} + rem = h_i$
8:     Check that $z_j + z_{j+1}2^b + \ldots + z_{j+B}2^{B*b} + rem = h_i$     ▷ Check that the samples are a decomposition of $h_i$
9:     **for** $k \in \{1, 2, 3, \ldots, B\}$ **do**
10:         Check that $0 \leq z_{i+k} < 2^B$
11:     **end for**
12:     Check that $0 \leq rem < 2^B$
13:     $j \leftarrow j + B$
14: **end for**
15: return $z$

---

---

**Algorithm 13** ZK_GAUSSIAN_SAMPLE

---

1: **Input:** Poseidon hashes $h$, sampling bit-width $b$
2: **Output:** Gaussian samples $z$
3: $z =$ ZK_UNIFORM_SAMPLE$(h, b)$
4: $z =$ LOOKUP_GAUSSIAN_INVERSE_CDF$(z)$
5: return $z$

---

## A.1. Security of *ExpProof*

**Completeness**   $\forall$ ZK_LIME configurations $cc$, input points $x$, and model weights $\mathbf{W}$

$$\Pr\left[\begin{array}{c} \mathsf{pp} \leftarrow \textit{ExpProof}.\mathsf{Setup}(1^k) \\ (pk, vk) \leftarrow \textit{ExpProof}.\mathsf{KeyGen}(pp) \\ (\mathsf{com}_{\mathbf{W}}, \mathsf{com}_r) \leftarrow \textit{ExpProof}.\mathsf{Commit}(pp, \mathbf{W}, r) \\ (o, e, \pi) \leftarrow \textit{ExpProof}.\mathsf{Prove}(pp, pk, cc, x, \mathsf{com}_{\mathbf{W}}, \mathbf{W}, \mathsf{com}_r, r_p, r_v) \\ \textit{ExpProof}.\mathsf{Verify}(pp, vk, cc, x, o, e, \mathsf{com}_{\mathbf{W}}, \mathsf{com}_r, \pi) = 1 \end{array}\right] = 1.$$

*Proof Sketch.* **Completeness.** The completeness proof mostly follows from the completeness of the underlying proof system (in our case, Halo2). We must also show that for any set of parameters there exists a LIME solution $\hat{w}$ and a feasible dual solution $v$ such that the dual gap between $\hat{w}$ and $\hat{v}$ is less than $\epsilon$. We know from the strong duality of Lasso that there exists a solution $w*$ and $v*$ such that the dual gap is 0 for any input points and labels, therefore such a solution exists. However, we also note that the circuit operates on fixed-point, discrete values (not real numbers), and it is not necessarily true that there are valid solutions in fixed-point. To solve this, the prover can use a larger number of fractional bits until the approximation is precise enough. $\qquad\square$

**Knowledge-Soundness**   We define the relation $\mathcal{R}_{lime}$ as:

$$\mathcal{R}_{lime} = \left\{ (cc, x, o, e, r_v, \mathsf{com}_{\mathbf{W}}, \mathsf{com}_r; \mathbf{W}, r_p, y, h, \hat{w}, \hat{v}) \left| \begin{array}{c} \mathsf{com}_{\mathbf{W}} = Com(\mathbf{W}) \\ \mathsf{com}_r = Com(\mathbf{r_p}) \\ o \leftarrow cc.f(\mathbf{W}, x) \\ h_i = \mathrm{Poseidon}(r_p + r_v, i) \\ z \leftarrow \mathrm{SAMPLE\_AROUND}(x, cc.\sigma) \\ y \leftarrow cc.f(\mathbf{W}, z) \\ \pi \leftarrow \mathrm{LIME\_KERNEL}(x) \\ y = cc.f(\mathbf{W}, z) \\ z' \leftarrow \sqrt{\pi} \times z \\ y' \leftarrow \sqrt{\pi} \times y \\ p \leftarrow \frac{1}{2n}\|y' - b - \hat{w}^T z'\|^2 + cc.\alpha\|\hat{w}\|_1 \\ d \leftarrow \frac{-n}{2}\|\hat{v}\|^2 + \hat{v}^T(y' - b) \\ p - d \leq cc.\epsilon \\ f \leftarrow (X^T)v \\ -cc.\alpha \leq f \leq cc.\alpha \end{array} \right. \right\}$$

There exists an extractor $\mathcal{E}$ such that for all probabilistic polynomial time provers $\mathcal{P}*$

$$\Pr\left[\begin{array}{c} \mathsf{pp} \leftarrow \textit{ExpProof}.\mathsf{Setup}(1^k) \\ (pk, vk) \leftarrow \textit{ExpProof}.\mathsf{KeyGen}(pp) \\ (cc, x, o, e, r_v, \mathsf{com}_{\mathbf{W}}, \mathsf{com}_r, \pi) \leftarrow \mathcal{P}(1^\lambda, \mathsf{pk}) \\ \textit{ExpProof}.\mathsf{Verify}(pp, vk, cc, x, o, e, r_v, \mathsf{com}_{\mathbf{W}}, \mathsf{com}_r) = 1 \\ (\mathbf{W}, r_p, y, h, \hat{w}, \hat{v}) \leftarrow \mathcal{E}^P(...) \\ (cc, x, o, e, r_v, \mathsf{com}_{\mathbf{W}}, \mathsf{com}_r; \mathbf{W}, r_p, y, h, \hat{w}, \hat{v}) \notin \mathcal{R}_{lime} \end{array}\right] \leq negl(\lambda).$$

*Proof Sketch.* **Knowledge Soundness.** Knowledge-soundness follows directly from the knowledge-soundness of the underlying proof system Halo2. The extractor runs the Halo2 extractor and outputs the Halo2 witness. By the construction of the circuit ZK_LIME, the extracted Halo2 witness satisfies the $\mathcal{R}_{lime}$ relation. $\qquad\square$

**Zero-Knowledge**   We say a protocol $\Pi$ is *zero-knolwedge* if there exists a polynomial time, randomized simulator $\mathcal{S}$ such that for all $(pk, vk) = \mathsf{Setup}(pp)$, for all $(x, w) \in \mathcal{R}$, for all verifiers $V$

$$\{P(pk, x, w)\} \approx \{S(pk, x)\}$$

*Proof Sketch.* **Zero-Knowledge.** Let the simulator $\mathcal{S}$ be the Halo2 simulator. For any $(cc, x, o, e, r_v, \mathsf{com}_\mathbf{W}, \mathsf{com}_r, \mathbf{W}, r_p, y, h, \hat{w}, \hat{v}) \in \mathcal{R}_{lime}$, we know that

$$\{\textit{ExpProof.Prove}(cc, x, o, e, r_v; \mathbf{W}, r_p, y, h, \hat{w}, \hat{v})\} \approx \{\mathcal{S}(cc, x, o, e, r_v)\}$$

by zero-knowledge of Halo2. $\square$

### A.2. LASSO Primal and Dual

Notation: Let $X \in \mathbf{R}^{n \times m}$ denote the data inputs, $y \in \mathbf{R}^n$ denote the labels and $\alpha > 0$ denote the regularization parameter or the LASSO constant. Let $w \in \mathbf{R}^m$ denote the primal variable and $v \in \mathbf{R}^n$ denote the dual variable.

The primal LASSO objective is given as, $l(w) = \frac{1}{2n}\|Xw - y\|_2^2 + \alpha\|w\|_1$ while the dual objective function is given as $g(v) = -\frac{n}{2}\|v\|_2^2 - v^\top y$ with the feasibility constraint $0 \leq L_\infty\left(X^\top v\right) \leq \alpha$ (Kim et al., 2007).

From a LASSO primal feasible $w$, it is possible to compute a dual feasible $v$ as (Kim et al., 2007):

$v = 2s(Xw - y)$

$s = \min\left\{\alpha / \mid 2\left(\left(W^T W x\right)_i - 2y_i\right)\mid \mid i = 1, \ldots, n\right\}$

We find that this dual is close enough to the dual optimal to get a good duality gap, however, in the worst case it is possible to apply traditional optimization methods to find a dual feasible with a smaller duality gap.

**Note on the duality gap threshold, $\epsilon$.** The duality gap condition is a stopping condition commonly used in optimization libraries. Therefore from an algorithmic perspective, is a 'parameter' fixed by the user based on how much error the user can tolerate. In our case the 'user' is actually the verifier/customer as the LASSO solution is the explanation for the verifier. As such, this threshold should be public (as is assumed in *ExpProof*) otherwise the prover can set it to arbitrary values. This value should ideally be provided by the verifier or set by regulators based on how much error is tolerable (independently of the model weights or the dataset).

Additionally, note that the duality gap condition $l(w) - g(v) \leq \epsilon$ implies that the difference between primal optimal and primal feasible values is bounded by $\epsilon$ as well, that is $l(w) - l(w^*) \leq \epsilon$. This in turn implies that the difference between the true explanation ($w^*$) and approximate explanation ($w$) is bounded as follows: $\|w - w^*\|_2 = O\left(\frac{\sqrt{n\epsilon}}{\lambda_{\min_+}(X)}\right)$, where X is the samples from the neighborhood and should be full rank and $\lambda_{\min_+}(X)$ is the smallest non-zero positive singular value of X.

# B. Experimental Details

## B.1. NN results

Next in Fig.5 we show results for using uniform sampling in the 'prediction similarity' evaluation, keeping rest of the parameters same. We observe similar numbers as before with very slight differences from Fig.2.

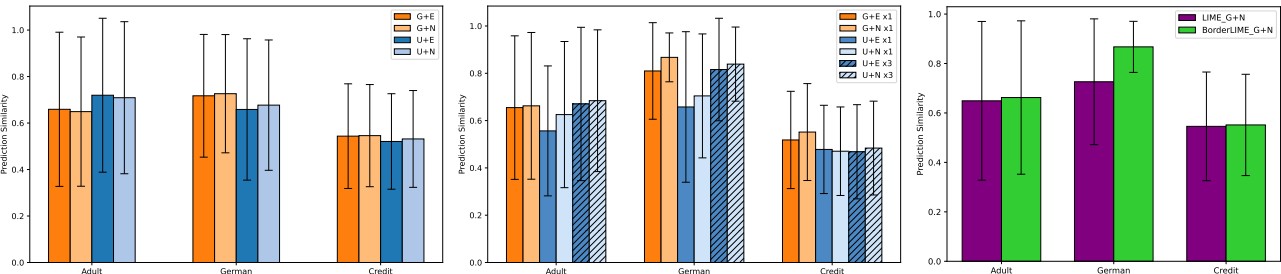

*Figure 5.* Results for NNs for $n = 300$ neighboring points and uniform sampling in the evaluation. Left: Fidelity of different variants of Standard LIME, Mid: Fidelity of different variants of BorderLIME, Right: Fidelity of Standard vs. BorderLIME.

Next we increase the neighborhood size, $n$, in LIME from 300 to 5000 samples and present the results in Fig.6. As expected the fidelity increases across the board due to better model fitting with a larger number of points. The size of the error bars only reduces significantly for the German dataset, showing that for more input points the explanations are faithful to the original decision boundaries. Additionally, the German dataset also has the highest fidelity explanations. Both these points hints towards smoother or more well-behaved decision boundaries learnt using the German dataset. Furthermore, BorderLIME consistently outperforms standard LIME pointing to better explanations.

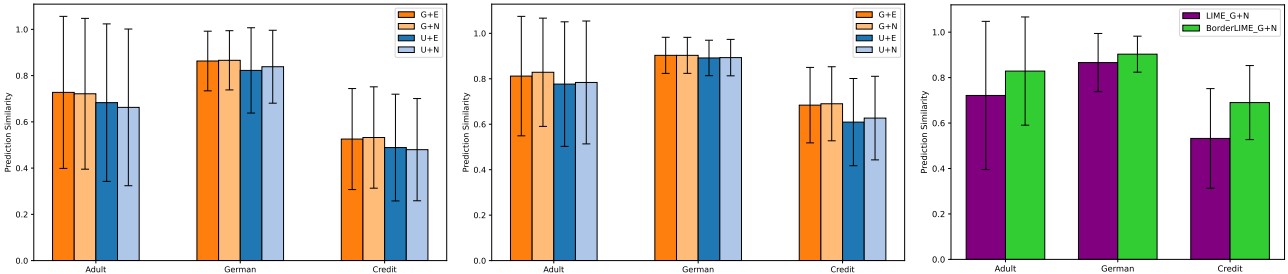

*Figure 6.* Results for NNs for $n = 5000$ neighboring points and gaussian sampling in the evaluation. Left: Fidelity of different variants of Standard LIME, Mid: Fidelity of different variants of BorderLIME , Right: Fidelity of Standard vs. BorderLIME.

## B.2. RF results

Next in Fig.7 and Fig.8 we show results for fidelity of explanations for Random Forests using gaussian and uniform sampling in the 'prediction similarity' evaluation respectively, keeping rest of the parameters same. We observe results as for NNs.

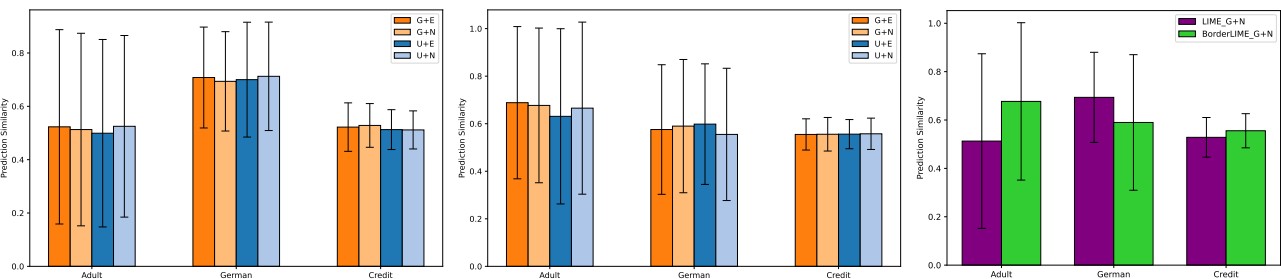

*Figure 7.* Results for RFs for $n = 300$ neighboring points and gaussian sampling in the evaluation. Left: Fidelity of different variants of Standard LIME, Mid: Fidelity of different variants of BorderLIME , Right: Fidelity of Standard vs. BorderLIME.

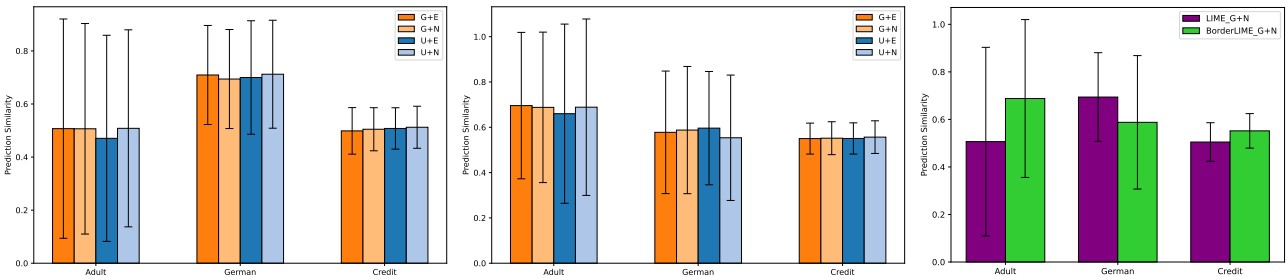

*Figure 8.* Results for RFs for $n = 300$ neighboring points and uniform sampling in the evaluation. Left: Fidelity of different variants of Standard LIME, Mid: Fidelity of different variants of BorderLIME , Right: Fidelity of Standard vs. BorderLIME.

Next we show the ZKP overheads for RFs in Fig.9 and Table 3. Trends and observations are similar as for NNs.

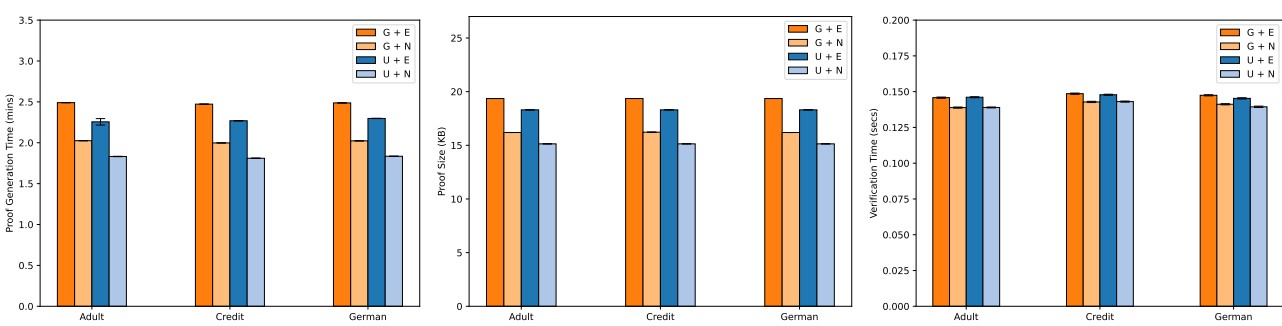

*Figure 9.* Results for RFs for $n = 300$ neighboring points. Left: Proof Generation Time (in mins), Mid: Proof Size (in KBs), Right: Verification times (in secs) for different variants of Standard LIME. All configurations use the same number of Halo2 rows, $2^{18}$, and lookup tables of size 200k.

| ZKP Overhead Type | BorderLIME | LIME |
|---|---|---|
| Proof Generation Time (mins) | $8.46 \pm 10^{-1}$ | $2.02 \pm 10^{-2}$ |
| Verification Time (secs) | $0.44 \pm 10^{-2}$ | $0.14 \pm 10^{-3}$ |
| Proof Size (KB) | $17.25 \pm 0$ | $16.20 \pm 10^{-2}$ |

*Table 3.* ZKP Overhead of BorderLIME and Standard LIME (both G+N variant) for RFs for 300 neighboring points. Overhead for BorderLIME is larger than that for LIME. Results are consistent across all datasets.

Next we increase the neighborhood size, $n$, in LIME from 300 to 5000 samples and present the results in Fig.10. As expected the fidelity increases across the board due to better model fitting with a larger number of points. We see slight reduction in error bars for German dataset. BorderLIME equals or outperforms standard LIME.

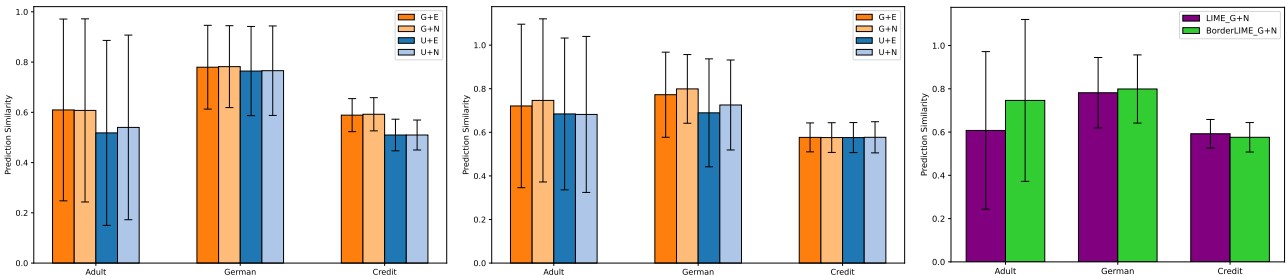

*Figure 10.* Results for RFs for neighboring points $n = 5000$ and gaussian sampling in the evaluation. Left: Fidelity of different variants of Standard LIME, Mid: Fidelity of different variants of BorderLIME , Right: Fidelity of Standard vs. BorderLIME.

