# OpenReview forum: "ExpProof : Operationalizing Explanations for Confidential Models with ZKPs"
_ICML.cc/2025/Conference — ICML 2025 poster_

### Official Review · Reviewer_H1H2 · 2025-02-27

**Overall Recommendation:** 1

**Summary:**

The paper introduces a novel cryptographic framework aimed at providing verifiable explanations for machine learning models while ensuring model confidentiality. The central idea is to leverage ZKPs and cryptographic commitments to guarantee the correctness of explanations in adversarial settings, where parties may have misaligned interests and could manipulate the explanations to their advantage. The authors propose a solution called ExpProof, which uses cryptographic commitments to bind a model to a fixed set of weights and explanation parameters, and ZKPs to prove that the explanations are computed correctly using the predefined explanation algorithm, all without revealing sensitive model details. The explanation algorithm is extended from LIME to overcome the significant computational overhead imposed by ZKPs. The paper includes a comprehensive set of experiments on NN and RF using standard datasets.

**Claims And Evidence:**

1. In "Solution Desiderata," I believe Model Uniformity and Model Consistency should be merged as one desiderata, as it is essentially achieved by the same cryptographic commitment and their objectives are quite similar. For "Model Confidentiality," current claim is misleading. The claim "the model $f$ is kept confidential" somewhat implies that the architecture of the model is also private, which is not the case in Alg 8.
2. The overall ZK_LIME looks very vague. First, what does it mean by "Generate proof Π of the above computation"? Do you mean something like proof aggregation or Nova's proof folding scheme that packs all the sub-proofs in the algorithm? Is Π a single folded proof or a bunch of proofs in this algorithm?
3. Alg 9 is also vague to me. In particular, how is the input parameter maximum dual gap $\epsilon$ determined? If I understand correctly, $\epsilon$ is data-dependent and would releasing $\epsilon$ as a public input undermine the confidentiality? And, the proof to the system's zero-knowledge property does not hold if $\epsilon$ is public and data/task-dependent.

**Essential References Not Discussed:**

Verifiable evaluations of machine learning models using zkSNARKs (preprint) -> the paper that describes ezkl, the framework on which it extensively relies.

Trustless Audits without Revealing Data or Models (ICML 2024)

**Experimental Designs Or Analyses:**

Please see my comments in "Methods And Evaluation Criteria" section.

**Methods And Evaluation Criteria:**

1. One of the biggest concerns is scalability. LIME is expected to explain moderately complex models that are difficult to analyze or interpret. However, the model used in the evaluation is too small, with only two layers, making it hard to justify its real-world usefulness. While ZKP introduces significant computational overhead, please at least consider using a model comparable to those studied in the ZKML paper.
2. Conceptually, the core contribution is unclear. My impression is that the paper resembles a technical report rather than a principled research contribution, as the proposed technique is largely an operational combination of existing solutions, particularly ezkl. For example, this pipeline could also be achieved through a combination of “Trustless Audits without Revealing Data or Models” (ICML 2024) and “ZKML: An Optimizing System for ML Inference in Zero-Knowledge Proofs” (EuroSys 2024), both of which provide ZK ML inference and training features. I encourage the authors to clearly articulate the conceptual novelty beyond a mere operational description of the process.

**Other Comments Or Suggestions:**

N/A

**Other Strengths And Weaknesses:**

I enjoyed reading the first two sections of the paper, which clearly present the motivation and overall narrative. However, the presentation could be improved by visualizing Algorithms 5 and 6 as a schematic diagram to help readers from the ML community better understand what is happening behind the scenes.

I also encourage the authors to strengthen (or articulate, if I missed something) the conceptual novelty and empirical usefulness of the proposed framework.

**Questions For Authors:**

I believe the paper would benefit from a major revision to address my concerns listed above, and the required workload may go beyond what is feasible during the rebuttal phase. Therefore, I do not have any explicit questions, though I am open to any clarifications.

**Relation To Broader Scientific Literature:**

The key contribution is the adaptation and encoding of the LIME algorithm into a zero-knowledge circuit. The core process involves (1) ZK ML inference, (2) ZK point sampling, and (3) ZK ML training. However, given that both (1) and (3) have already been addressed by existing solutions, and (2) is a standard cryptographic trick, the overall contribution to the broader community appears weak.

**Theoretical Claims:**

Please see my comments in "Claims And Evidence" section.

---

> ### Author Rebuttal · Authors · 2025-04-01
>
> We appreciate the insightful comments and the time taken by the reviewer to review our paper. We are glad that the reviewer enjoyed reading the first two sections of the paper clearly presenting the motivation and overall narrative.
>
> Major:
>
> - **Missing References**: Thank you for sending these interesting papers our way, we weren’t aware of them. We’ll cite these in our related work section. As future work, it would be interesting to see if some of the systems level insights from these papers can lead to improvements in our ZKP overheads.
>
> - **Contribution**: As the ZKP & ML communities are very disjoint, *the key contribution of our paper is to identify a long-standing problem in explainable machine learning which the community wanted to solve, but couldn’t solve with traditional methods [1,2], and then solve it with ZKPs* – without our solution explanations cannot be used in adversarial contexts. We reveal the utility of ZKPs in operationalizing explanations to the *explainability community, which is mostly unaware of the existence of this primitive or has not made the connection we make*. As we see, **our contribution is appreciated by members of XAI community: quoting reviewer 96NK “The paper introduces a new paradigm for explanations” & “This is an interesting contribution to the explainable AI community”; quoting Reviewer JYN6 “The proposed problem is interesting” & “ExpProof is a theoretically sound solution, with many benefits”**.
>
>     *We don't claim to make original contributions to the ZKP technology, the main contribution here is an ML contribution*. But using existing technology, together with some smart tricks, *we provide a realistic (not-toy, written in ZKP libraries that are used in practice) working implementation of zk_LIME as a start point to the explainability community*.
>
> - **Scalability** : The focus of our work is societal applications (such as finance, health, justice) where the “right to explanation” is applicable and misalignment of interests occurs organically. In many of these use-cases the data is tabular & the most popular choice for tabular data is still small neural networks and random forests. *These models are enough to achieve SOTA accuracy* [3, 4, 5]. In such domains we do see the usefulness of our experiments done on these class of models and popular tabular datasets. Additionally, as the first work in using ZKPs for explanations and giving a unique solution to a  long-standing problem, we hope future work can address the scalability issue.
>
>     Having said this, we conduct some experiments/ablations which can be found at https://anonymous.4open.science/r/expproof_experiments_rebuttal-6C75/experiments_rebuttal_expproof_icml.pdf. We find that the maximum time in zk_LIME is taken for proving inferences and the time for this dominates others as the model size grows. *Since expproof treats the inference proof library as a plug-and-play module, improvements in inference proofs (which is a very active area of research) will directly translate in our tool*.
>
>
> Minor:
>
> - **Architecture** : Thanks for catching this. We do mention in L222-223-right-side that architecture is public, but we will make necessary amends as suggested by you in the paper to emphasize this.
>
> - **Proof generation** : We do not use proof aggregation or proof folding, it is a monolithic proof. When we say "Generate proof Π of the above computation", we mean that we encode the above computation as a Halo2 relation, and use Halo2 to prove its correctness as a monolithic proof. Each sub-routine we call does not generate a separate proof, it is simply a sub-computation in the Halo2 relation, and we only split it up for organization purposes. We will highlight this in the paper.
>
> - **$\epsilon$ in duality gap** : The duality gap condition is a stopping condition commonly used in optimization libraries (since it implies f(x)-f* <= $\epsilon$ where f is the primal). Therefore from an algorithmic pov, $\epsilon$ is a ‘parameter’ fixed by the user based on how much error it can tolerate. In our case the ‘user’ is actually the verifier/customer as the lasso solution is the explanation for the verifier. As such, this threshold should be public (as is assumed in ExpProof) otherwise the prover can set it to anything. This value should ideally be provided by the verifier or set by regulators based on how much error is tolerable (independently of the model weights or the dataset). We use $\epsilon$ = 0.001. We will highlight this in the paper.
>
> Please let us know if you have any remaining concerns!
>
> Authors
>
> [1] Auditing local explanations is hard. Bhattacharjee et. al
>
> [2] Post-hoc explanations fail to achieve their purpose in adversarial contexts. Bordt et.al.
>
> [3] Individual Fairness Guarantees for Neural Networks. Benussi et.al.
>
> [4] Well-tuned Simple Nets Excel on Tabular Datasets. Kadra et.al.
>
> [5] Better by Default: Strong Pre-Tuned MLPs and Boosted Trees on Tabular Data. Rabuchev et. al.

---

> > ### Comment · Reviewer_H1H2 · 2025-04-06
> >
> > Thank you for the new experiments and clarification. However, my concerns about the scalability of the method remain. The new models still do not match those used in the ZKML paper. Additionally, the novelty of the work is still a concern. From my understanding, it is a combination of several standard cryptographic tools. Even when considered as an ML paper, its unique contribution to the ML community remains unclear, as similar results could be achieved using a combination of existing tools developed within the community (e.g., the ICML paper I mentioned). Therefore, I have decided to maintain my current rating.

---

> > > ### Author Response · Authors · 2025-04-06
> > >
> > > Thank you for your comments.
> > >
> > > - **Contribution** : how will the Explainable AI (XAI) community know that their long-standing problem can be resolved by ZKPs if they don’t know that such a thing exists? or have not made this connection? As mentioned before, the XAI and ZKP communities are very disjoint and therefore, the connection that explanations can be operationalized in adversarial contexts with ZKPs is not prevalent & evident to the XAI community.
> > >
> > >     **Showing to the XAI community that their long-standing problem can be resolved with ZKPs and providing a working realistic prototype is our key contribution and as mentioned earlier it is appreciated by the other reviewers. On top of this, we also show how popular XAI algorithms (LIME and BorderLIME) can be made more ZKP-amenable -- it is not just about picking up an algorithm and blindly implementing it in ZKP as-is**.
> > >
> > > To reiterate, ours is an ML contribution, not a ZKP contribution and is extremely relevant to the XAI community.
> > >
> > >
> > > - **Scalability** : As is evident from new experiments, the bottleneck of the method is inference proofs for the sampled neighborhood. But fortunately zk_LIME treats the inference proof part as a plug-and-play module and therefore advances in inference proofs (which is a very active research area) will directly translate to our tool. **This is true for models of any scale. Additionally, we believe due to focus on societal applications where tabular data and small NNs, Random forests achieve SOTA, our experiments are valuable.**
> > >
> > > Authors

---

### Official Review · Reviewer_JYN6 · 2025-03-13

**Overall Recommendation:** 4

**Summary:**

The paper proposes to compute model explanations, in particular LIME, using Zero Knowledge Proofs (ZKP). This allows consumers (users) of a service to receive verifiable explanations for their predictions, without the service having to reveal their model and thus preserving their IP. The paper theoretically constructs the ZKP protocol and experimentally evaluates their feasibility.

**Claims And Evidence:**

Yes, the claims are supported both theoretically, as well as empirically. In particular, the authors show that the method guarantees (1) model uniformity, (2) explanation correctness, (3) model consistency, (4) model confidentiality, and (5) technique reliability. The experiments demonstrate that the method is feasible for very small models (Random Forests, NN with 2 layers and 16 hidden units) and datasets.

**Essential References Not Discussed:**

See above

**Experimental Designs Or Analyses:**

The experiments compare the prediction similarity between the explanation and the original model, and the computational time to arrive at those explanations using ZKPs. The experiments make sense, although it would have be helpful to investigate how the method scales to larger models and datasets.

**Methods And Evaluation Criteria:**

The method makes sense theoretically, and solves the proposed problem. However, as common with ZKPs, the computational complexity and the amount of data communicated is significant, and therefore only practical for very small models. The evaluation demonstrates this on three simple datasets. It would be interesting to also show how the computation scales with larger models and more complex data.
The evaluation is limited to 50 samples from the test sets, and it is unclear how these are selected. It would be good to expand this evaluation to at least 100, ideally more data-points and ensure random, uniform sampling.

**Other Comments Or Suggestions:**

- Algorithm 5 is the main contribution of the work; it should imho be part of the main paper and not the appendix. If space is a concern, I suggest to defer Algorithms 2-5 to the appendix instead.
- The hyper-parameter $\epsilon$ is crucial, but I did not find what value is used for experiments.
- The claim that “explanations […] are often obligated by regulations” needs better support. A reference to a Wikipedia article explaining the term is not sufficient. Either cite corresponding legislation, or remove the claim.
- There should be a space inserted after all instances of the *ExpProof* name.

**Other Strengths And Weaknesses:**

**Strengths**

- Sections 1-5 of the paper are very well written and easy to follow. I especially liked the consistent example in the introduction.
- The proposed problem is interesting and novel, and looks at explanation in a different context.
- ExpProof is a theoretically sound solution, with many benefits. It preserves the IP of the service provider, gives cryptographic guarantees, and does not require a trusted third party for verification.

**Weaknesses**

- Due to the substantial computational overhead, the practical applicability is limited.
- The paper is poorly positioned in the related scientific literature (see above)
- No code is provided for review

**Questions For Authors:**

1. What is the value of hyper-parameter $\epsilon$?
2. How are the 50 samples from the dataset selected?
3. What do you mean by “no explicit parallelization”? Does this mean there is “implicit” parallelization? What does this mean exactly?

**Relation To Broader Scientific Literature:**

The paper is very poorly positioned in the broader scientific literature. The related works section is overly brief, and only mentions some references from the ZKP literature. However, this paper combines several different areas: (1) ZKPs, (2) explainability methods, (3) adversarial attacks. (2) and (3) are missing entirely.

**Theoretical Claims:**

Yes, I checked the correctness of the proof in 5 and app A1. It looks correct to me. however, I am not a cryptographer and therefore lack the expertise to detect subtle issues with SKPs.

One issue I would like to see discussed more: the authors provide a ZKP proof that verifies the solution to the LIME optimization problem, instead of the computation of the solution. This is a neat trick to safe computational resources. However, there are implications that need to be verified. In particular, there may be multiple valid solutions to the optimization problem, especially when we allow an $\epsilon$-gap to the optimum. This could be exploited by the service to provide a different solution than the one computed by the LIME algorithm. The authors should
(1) prove and verify that all such solutions are valid explanations
(2) adjust the phrasing of the guarantees to indicate that the ZKP does not check for correct LIME solutions, but only verify those.

---

> ### Author Rebuttal · Authors · 2025-04-01
>
> We appreciate the insightful comments and the time taken by the reviewer to review our paper. We are glad that the reviewer finds our paper well-written, easy to follow, thinks the proposed problem in our paper is interesting and novel and finds our cryptographic solution theoretically sound with many benefits.
>
> Next we address your concerns and questions.
>
>
> - **Related works** : Thanks for pointing this out. While we did cite the closely related work on adversarial attacks for explanations in the introduction and other sections of the paper, it is a good idea to dedicatedly talk about these and other related papers on explainability and adversarial attacks in the related work section. We will make this change in the paper.
>
> - **Evaluation** : The test points were *randomly* sampled from the respective test sets of the datasets (mentioned in L321right). The ZKP results do not change by increasing the sample size as the variance is extremely small.
>
> - **$\epsilon$ value** : We use 0.001. We will add this value to the paper.
>
> - **Parallelization** : The ZKP library we use, ezkl, automatically does multithreading on all the available cores. By “explicit” we mean that other than ezkl’s multithreading, we do not use gpus, do not modify ezkl to do more parallelization and do not do any of the steps in zk_LIME in parallel by ourselves. We will clarify this in the paper.
>
> - **Scalability** : The focus of our work is societal applications (such as finance, health, justice) where the “right to explanation” is applicable and misalignment of interests occurs organically. In many of these use-cases the data is tabular and the most popular choice for tabular data is still small neural networks and random forests. *These models are enough to achieve SOTA accuracy* [1,2,3]. In such domains we do see the usefulness of our experiments done on these class of models and popular tabular datasets. Additionally, *as the first work in using ZKPs for explanations and giving a unique solution to a  long-standing problem, we hope future work can address the scalability issue.*
>
>     Having said this, we conduct some experiments/ablations which can be found at https://anonymous.4open.science/r/expproof_experiments_rebuttal-6C75/experiments_rebuttal_expproof_icml.pdf. We find that the maximum time in zk_LIME is taken for proving inferences and the time for this dominates others as the model size grows. Since expproof treats the inference proof library as a plug-and-play module, improvements in inference proofs (which is a very active area of research) will directly translate in our tool.
>
>     Dimensionality doesn’t affect the time as much, as is also evident in Fig. 3 in our paper where the ZKP overhead is the same across datasets of different dimensions. This is because dimensionality plays the most role in LASSO and sampling checks and the proving time for these is highly overshadowed by inference proof times.
>
>
>
> Please let us know if you have any remaining concerns. We look forward to hearing from you!
>
> Authors
> ***********
>
> [1] Individual Fairness Guarantees for Neural Networks. Benussi et.al. 2022
>
> [2] Well-tuned Simple Nets Excel on Tabular Datasets. Kadra et.al. 2021
>
> [3] Better by Default: Strong Pre-Tuned MLPs and Boosted Trees on Tabular Data. Rabuchev et. al. 2024

---

> > ### Comment · Reviewer_JYN6 · 2025-04-04
> >
> > Thank you for your reply to my questions and the clarifications.
> >
> > **Scalability**: Thank you for the additional evaluations. I agree that there are some applications where it makes sense, and this being the first work providing ZKP guarantees the value stands even if direct real-world applications are limited. I therefore see it as a limitation of the method but not a reason against acceptance.
> >
> > Could you please also reply to my comment under "Theoretical Claims"?
> >
> > > One issue I would like to see discussed more: the authors provide a ZKP proof that verifies the solution to the LIME optimization problem, instead of the computation of the solution. This is a neat trick to safe computational resources. However, there are implications that need to be verified. In particular, there may be multiple valid solutions to the optimization problem, especially when we allow an -gap to the optimum. This could be exploited by the service to provide a different solution than the one computed by the LIME algorithm. The authors should (1) prove and verify that all such solutions are valid explanations (2) adjust the phrasing of the guarantees to indicate that the ZKP does not check for correct LIME solutions, but only verify those.

---

> > > ### Author Response · Authors · 2025-04-04
> > >
> > > Dear Reviewer,
> > >
> > > Thank you so much for your engagement, we deeply appreciate it. Following is our reply that you asked for.
> > >
> > > ************
> > > Duality gap clarification : This is indeed an important point, thanks for raising it – we will be including more discussion on this in the paper, as suggested by the reviewer.
> > >
> > > Using the duality gap condition for verification saves time but implies that there could be multiple solutions that satisfy this condition, *zk_LIME proves that the given explanation is one of those solutions* (we will highlight this in the paper and adjust the phrasings as suggested by you).
> > >
> > > “prove and verify that all such solutions are valid explanations” : not sure what the reviewer exactly means by this, but here is our best attempt – it is not possible to enumerate all $\epsilon$-gap solutions because of continuous features. However, since we have an $\epsilon$ duality gap, it constraints the explanations from being arbitrarily far from the true explanation. Some theoretical bounds can be given here : using the duality gap condition primal and dual are bounded $l(w) - g(v) <= \epsilon$, which implies that the approximate primal and true primal values are bounded $l(w) - l(w*) <= \epsilon$, which in turn implies that the difference between the true (w*) and approximate explanation/lasso-solution (w) is bounded -- $L2(w,w*) <= O(sqrt(\epsilon)/\lambda_{min+}(X))$ where X is the samples from the neighborhood and should be full rank, $\lambda_{min+}(X)$ is the smallest non-zero positive singular value of X. (happy to add this to the paper)
> > >
> > >
> > >
> > > Additionally, we would like to highlight that the duality gap condition is a stopping condition commonly used in optimization libraries.Therefore from an algorithmic pov, $\epsilon$ is a parameter fixed by the user based on how much error it can tolerate. For our case the user is the verifier and therefore, the $\epsilon$ value should ideally be provided by the verifier or set by regulators based on how much error is tolerable or just set to the default values used by popular optimization libraries. In *zk_LIME $\epsilon$ is a public parameter (not a private information) which allows for this flexibility. The $\epsilon$ value we use is 0.001.*
> > >
> > > ************
> > >
> > > Please let us know if there are any more concerns. *If you have no more concerns, we kindly request you to consider raising your score, we will deeply appreciate your support for the paper.*
> > >
> > > Thanks,
> > >
> > > Authors

---

### Official Review · Reviewer_ormk · 2025-03-13

**Overall Recommendation:** 3

**Summary:**

This paper proposes a solution for operationalizing explanations in adversarial contexts where the involved parties have misaligned interests. The authors focus on LIME and propose a method called ExpProof, which integrates ZKPs to ensure that explanations remain trustworthy while maintaining model confidentiality. The authors explore different sampling strategies (Gaussian vs. Uniform) and kernel choices (Exponential vs. None) to balance explanation fidelity and computational efficiency in a ZKP setting. Experiments on three datasets demonstrate the feasibility of ExpProof for both Neural Networks and Random Forests.

## update after rebuttal

I have decided to maintain my current (positive) score because the authors have addressed all my concerns. After reading the other reviewers' comments, I agree with some of them and believe this work does have certain limitations, which may somewhat impact its value. So I am keeping a weak accept (3) rather than upgrading to an accept (4).

**Claims And Evidence:**

Yes, I think the claims made in the submission are supported by clear and convincing evidence.

**Essential References Not Discussed:**

I am not familiar with this domain, so I am unsure whether any relevant papers may have been overlooked in the citations.

**Experimental Designs Or Analyses:**

Overall, the experimental design and analysis seem sound. The only aspect I am uncertain about is whether the neural networks used in the experiments are too simple (a two-layer fully connected network), considering that modern neural networks typically have a much larger number of parameters and more complex architectures. If the neural network were more complex, it is unclear whether LIME would still provide reliable explanations, potentially affecting the significance of the proposed approach.

**Methods And Evaluation Criteria:**

Yes, the proposed methods and evaluation criteria make sense for the problem at hand.

**Other Comments Or Suggestions:**

The paper focuses solely on adapting ZKP to work with LIME, without exploring its compatibility with other explanation methods such as SHAP. It remains unclear whether integrating ZKP with SHAP would require a significant redesign or if the proposed approach naturally extends to other explainability techniques. Given that this is the first work applying ZKP to explanations, it is acceptable for the paper to focus only on LIME. However, In my view, this limitation somewhat affects the overall contribution.

**Other Strengths And Weaknesses:**

The paper is well-written and easy to follow. The clear presentation of ideas made it accessible, even for someone less familiar with the domain. I learned a lot from reading it, and I appreciate the authors’ efforts in presenting their work so clearly.

**Questions For Authors:**

- From my understanding of LIME, the original version (G+E) should generally perform better than any of the other variants. However, in Figure 2, it seems that N generally performs better than E. Why does this happen?
- Isn’t the default version of LIME (G+E)? Why does Figure 2 (right) compare standard LIME and BorderLIME using G+N instead?
- BorderLIME does not seem to provide a significant advantage over standard LIME, as it only outperforms LIME on the German dataset. Given its substantial inefficiency -- its proof generation and verification times are both 3x longer than those of LIME, and its proof size is 1.8x larger -- it raises the question of whether BorderLIME is practically necessary. Would standard LIME be a more efficient and sufficient choice in most cases?

**Relation To Broader Scientific Literature:**

According to the authors’ claims, this is the first work to identify the need for proving explanations and to propose ZKP-based solutions for this purpose.

**Theoretical Claims:**

No, I didn't check them.

---

> ### Author Rebuttal · Authors · 2025-04-01
>
> We appreciate the insightful comments and the time taken by the reviewer to review our paper. We are very glad that the reviewer finds our paper well-written, easy to follow, accessible and could learn a lot from it – this is very rewarding!
>
> Next we address your concerns and questions.
>
> Major:
>
> - **SHAP** : From our understanding, KernelSHAP is very similar to LIME (the major difference is the kernel). We believe KernelSHAP can be implemented using our zk_LIME implementation with reasonably easy modifications. The kernel for KernelSHAP is simpler than that for LIME, in the sense that it does not include exponentials, just arithmetic operations.
>
> - **LIME for modern networks** : We agree with your concern that LIME may not be the best explanation for large scale models of today. Having agreed on this, we want to highlight that the focus of our work is societal applications (such as finance, health, justice) where the “right to explanation” is applicable and misalignment of interests occurs organically. In many of these use-cases the data is tabular and the most popular choice for tabular data is still small neural networks and random forests, where LIME is one of the popular XAI tools. Additionally, *as the first work in using ZKPs for explanations and giving a unique solution to a long-standing problem, we hope future work can look at modern models and their suitable explanations.*
>
> Minor:
>
> - **Why G+N** : G+N variant performs equally well as G+E mainly because the gaussian distribution captures the behavior of the exponential kernel in a way, by sampling more around the mean than far off. To answer your next question about why G+N results were compared instead of G+E – this is because G+N and G+E are almost equally faithful but using G+N would save on the ZKP costs (due to absence of exponential kernel). So any practical ZKP system would rather choose G+N over G+E.
>
> - **BorderLIME**: The goal of BorderLIME is to provide non-vacuous explanations when the input point is far from the decision boundary; as such the benefits will show only in cases when the input points are far from the decision boundary. In cases where most of the test points are close to the decision boundaries, borderlime isn’t designed to give additional benefits while in those where the points are far off from the decision boundaries, borderlime should give significant improvement in fidelity.
>
> Please let us know if you have any remaining concerns. We look forward to hearing from you!
>
> Authors
> ****************
> [1] Individual Fairness Guarantees for Neural Networks. Benussi et.al. 2022
>
> [2] Well-tuned Simple Nets Excel on Tabular Datasets. Kadra et.al. 2021
>
> [3] Better by Default: Strong Pre-Tuned MLPs and Boosted Trees on Tabular Data. Rabuchev et. al. 2024

---

### Official Review · Reviewer_Zm91 · 2025-03-14

**Overall Recommendation:** 2

**Summary:**

The paper proposes a protocol with a zero-knowledge proof to ensure that a provided explanation is correct while maintaining confidentiality of the model parameters. In particular, the paper focuses on LIME and standard ZKP libraries. This involves modifying the pipeline so that the protocol is computationally efficient. Finally, the paper provides experiments on the runtime and explanation fidelity of the method.

### update after rebuttal
I am maintaining the current score following the rebuttal. I am still not convince on the assumption regarding the model architecture being public which limited the practicality of the proposed method.

**Claims And Evidence:**

1. On the commitment phase: The paper states 'the model owner commits to a fixed set of model weights W belonging to the original model f.' It's not clear how this is implemented in practice. The approach assumes the model architecture is public, which may not be realistic as architecture details could leak sensitive information. Additionally, how large is this fixed set of model weights? While a larger set would enhance privacy, this raises questions about the practical management of such information.

2. While the introduction focuses on how 'post-hoc explanations are highly problematic in an adversarial context' and presents ExpProof as a solution, the experiments don't include testing ExpProof in these adversarial scenarios. This creates a gap between the paper's motivation and its empirical validation.

**Essential References Not Discussed:**

N/A

**Experimental Designs Or Analyses:**

N/A

**Methods And Evaluation Criteria:**

The paper evaluates the approach using running time and the fidelity of LIME. The runtime assessment is appropriate for a computational protocol. However, the fidelity of LIME appears to function more as an ablation study rather than the primary result. It might be helpful to consider adding a metric that shows how well this approach could protect against adversarial explanations, possibly by measuring the percentage of manipulations it can detect or prevent.

**Other Comments Or Suggestions:**

N/A

**Other Strengths And Weaknesses:**

N/A

**Questions For Authors:**

N/A

**Relation To Broader Scientific Literature:**

The contribution provides an approach to solve the problem of adversarial post-hoc explanations which is significant. However, some assumptions required could be too strong (see Claims and Evidence).

**Theoretical Claims:**

N/A

---

> ### Author Rebuttal · Authors · 2025-04-01
>
> We appreciate the insightful comments and the time taken by the reviewer to review our paper.
>
> Below we address your concerns.
>
> - **Commitment phase** : As mentioned in our paper, commitments are a standard procedure in cryptography and it is well known how to implement this in practice and preexists in ZKP libraries. Models with a few billion parameters can be committed to in under a minute.  In our implementation (following EZKL), we use KZG as a vector commitment to commit to the weights. The model owner uses cryptographic commitments (such as KZG,  Pedersen, SHA256 hashing) to commit to the exact model weights, which binds the model owner to the weights while not revealing them to the verifier.
>
> - **Architecture being public** : Regarding architecture details being public, currently this is the state of research in the ZKP community and is a standard assumption; we hope future research in ZKPs for ML can address this limitation. Alternatively, in certain domains standard or published SOTA architectures might be used commonly and hence this may not be extremely sensitive information, so revealing it in order to get verifiable explanations may be a good tradeoff.
>
> - **Evaluation in adversarial contexts** : There is perhaps a slight misunderstanding here. By adversarial, we mean when the model might be swapped by the model developer at any given point or the explanations might be crafted instead of being the real ones from the model (as mentioned in L38right-56left). These problems are eliminated by ZKPs + commitments because of their theoretical guarantees, it is not an empirical question.
>
> Please let us know if you have any remaining concerns. We look forward to hearing from you!
>
> Authors

---

### Official Review · Reviewer_96NK · 2025-03-17

**Overall Recommendation:** 4

**Summary:**

The paper introduces ExpProof, a system that produces model predictions, explanations for the prediction and proof that the explanations are correct without revealing the model's weights. The key idea is to use cryptographic commitments and zero-knowledge proofs to ensure that a model owner cannot cheat when providing an explanation (for example, to provide a crafted explanation for the prediction). The model owner commits to a fixed model and commits to the parameters of the explanation algorithm. Then, for each query input, the model owner returns the prediction, a LIME-based explanation, and a ZKP, indicating that this explanation truly comes from the committed model.

**Claims And Evidence:**

Yes, the claims made in the submission are clearly supported by evidence.

**Essential References Not Discussed:**

Does not apply.

**Experimental Designs Or Analyses:**

Yes, the soundness and validity of the experimental designs are sound.

**Methods And Evaluation Criteria:**

Yes, the proposed methods do make sense for the problem at hand.

**Other Comments Or Suggestions:**

Does not apply.

**Other Strengths And Weaknesses:**

Strengths:
* The paper introduces a new paradigm for explanations. It’s the first to integrate Zero-Knowledge Proofs with an explanation algorithm to prove the explanation is correct and uses the intended model. This is an interesting contribution to the explainable AI community and to applied cryptography.
* The experiments cover multiple variants of LIME, two model classes, and three datasets, giving confidence that the system works in different scenarios.

Weaknesses:
* The paper does not tackle situations where the model is adversarially trained to produce misleading explanations. If a malicious model owner trains a model to have plausible explanations for some outputs, the presented technique cannot help. ExpProof only works assuming the model was honestly trained and the behaviour and explanation presented are internally consistent.
* While the overhead is reasonable for small models, it may become a bottleneck for larger models. The experiments used a 2-layer neural network and a small random forest. It’s not evaluated on deeper networks or datasets with a bigger number of features.

**Questions For Authors:**

1. Could you elaborate on the exact adversarial threat model you assume? For instance, you discuss an adversarial model owner who might manipulate explanations. However, it does not cover the case where the model owner is adversarial and is manipulating the model during training. This could be better presented in the paper as well.
2. You focus on LIME due to its relative simplicity for ZKPs. Have you considered other standard explainability techniques like SHAP?
3. What are the main factors that limit the scalability of ExpProof to larger neural networks or more complex architectures?
4. LIME can sometimes struggle with meaningful local sampling in high-dimensional feature spaces. How does that interaction play out in ExpProof, and does the proof cost scale with dimensionality?

**Relation To Broader Scientific Literature:**

The paper positions itself well in the literature and discusses all relevant work that I am aware of.

**Theoretical Claims:**

Does not apply.

---

> ### Author Rebuttal · Authors · 2025-04-01
>
> We appreciate the insightful comments and the time taken by the reviewer to review our paper. We are glad that the reviewer thinks our paper introduces a new paradigm for explanations, acknowledges that we are the first to integrate ZKPs with an explanation algorithm and finds our paper an interesting contribution to the XAI and applied cryptography communities.
>
> Next we address your concerns and questions.
>
> - **SHAP** : From our understanding, KernelSHAP is very similar to LIME (the major difference is the kernel). We believe KernelSHAP can be implemented using our zk_LIME implementation with reasonably easy modifications. The kernel for KernelSHAP is simpler than that for LIME, in the sense that it does not include exponentials, just arithmetic operations.
>
> - **Threat Model** : The model developer having access to the training data trains the model honestly. The parameters as mentioned in L661-665 are public and are assumed to be set honestly. It doesn’t have information about the input queries it will see. When presented with an input query, it generates both the prediction and explanation, which could be manipulated by either changing the model prediction arbitrarily, using a different model than the one it trained or using a different algorithm to generate the explanation. We will highlight this threat model in the paper.
>
> - **Factors limiting Scalability of ExpProof** : The main bottleneck is proving inferences for the sampled points (in the neighborhood of input). The total time taken for proving inferences increases as more points are sampled around the input point and as the models get more complex. However,  *since expproof treats the inference proof library as a plug-and-play module, improvements in inference proof times (which is a very active area of research) will directly translate in our tool*. We did some experiments to study the bottleneck, can be found here : https://anonymous.4open.science/r/expproof_experiments_rebuttal-6C75/experiments_rebuttal_expproof_icml.pdf.
>
> - **Scaling wrt dimensionality** : Dimensionality doesn’t affect the time as much, as is also evident in Fig. 3 in our paper where the ZKP overhead is the same across datasets of different dimensions. This is because dimensionality plays the most role in LASSO and sampling checks and the proving time for these is highly overshadowed by inference proof times.
>
> - **High dimension sampling** : From our knowledge, latin hypercube sampling (LHS) is seen to do better in high dimensions, this is also implemented in the python LIME library. *One way of implementing LHS in a ZKP library would be through a random shuffling module. Since ExpProof is very modular, this can be integrated into our tool once implemented*.
>
>
> Please let us know if you have any remaining concerns. We look forward to hearing from you!
>
> Authors

---

### Decision · Program_Chairs · 2025-05-01

**Decision:**

Accept (poster)

**Comment:**

This paper introduces ExpProof, a methodology to provide explanations that are certified. That is, we know that the explanations are correctly computed without looking at the weights of the model. This is achieved using Zero-Knowledge Proofs (ZKPs), a cryptographic primitive.

The reviewers see merit in the paper, in particular, the novelty of the method and the efforts to make it understandable to an audience not familiar with ZKPs. Their main concern, namely the scalability of the method was addressed in the rebuttal. Another concern was the limited scope of the paper, which is currently strongly focused on LIME. This focus seems reasonable for a proof-of-concept paper, and I am recommending the paper to be accepted.